# High-throughput sensitive screening of small molecule modulators of microexon alternative splicing using dual Nano and Firefly luciferase reporters

Andrew J. Best[1] ✉, Ulrich Braunschweig [1], Mingkun Wu[1,2], Shaghayegh Farhangmehr[1,2], Adrian Pasculescu[3], Justin J. Lim[1,2], Lim Caden Comsa[1,2], Mark Jen[3], Jenny Wang[3], Alessandro Datti [4], Jeffrey L. Wrana [2,3], Sabine P. Cordes [2,3], Rima Al-awar[5], Hong Han[1] & Benjamin J. Blencowe [1,2] ✉

Disruption of alternative splicing frequently causes or contributes to human diseases and disorders. Consequently, there is a need for efficient and sensitive reporter assays capable of screening chemical libraries for compounds with efficacy in modulating important splicing events. Here, we describe a screening workflow employing dual Nano and Firefly luciferase alternative splicing reporters that affords efficient, sensitive, and linear detection of small molecule responses. Applying this system to a screen of ~95,000 small molecules identified compounds that stimulate or repress the splicing of neuronal microexons, a class of alternative exons often disrupted in autism and activated in neuroendocrine cancers. One of these compounds rescues the splicing of several analyzed microexons in the cerebral cortex of an autism mouse model haploinsufficient for Srrm4, a major activator of brain microexons. We thus describe a broadly applicable high-throughput screening system for identifying candidate splicing therapeutics, and a resource of small molecule modulators of microexons with potential for further development in correcting aberrant splicing patterns linked to human disorders and disease.

Alternative splicing, the process by which exon and intron sequences are differentially included in transcripts, affects more than 90% of human genes and is frequently disrupted in diseases and disorders[1–4]. Approximately 50% of human genetic diseases are caused by mutations that affect cis-acting RNA sequences comprising a splicing code that governs the recognition and regulation of splice sites[5,6]. Additional diseases arise through mutations in—or altered regulation of—trans-acting factors that control the splicing

process. The enormous burden of splicing deficiencies on human health has stimulated intense efforts to develop therapeutics designed to specifically reverse deleterious splicing patterns, as well as to modulate splicing events that can correct disease-altered gene expression levels. Initial success has come from the development of antisense oligonucleotide (ASO) and small molecule-based therapeutics for the treatment of spinal muscular atrophy (SMA)[7–9]. These therapeutics promote splicing of exon 7 in Survival of Motor

[1]Donnelly Centre, University of Toronto, Toronto, ON, Canada. [2]Department of Molecular Genetics, University of Toronto, Toronto, ON, Canada. [3]Lunenfeld-Tanenbaum Research Institute, Mount Sinai Hospital, Toronto, ON, Canada. [4]Department of Agricultural, Food, and Environmental Sciences, University of Perugia, Perugia, Italy. [5]Department of Pharmacology and Toxicology, University of Toronto, Toronto, ON, Canada. ✉e-mail: andy.best@utoronto.ca; b.blencowe@utoronto.ca

Neuron 2 (SMN2) transcripts, thereby increasing the production of functional SMN2 protein, which compensates for the loss of SMN1 as a consequence of mutations in SMA patients. While these efforts demonstrate the major potential of splicing-directed therapeutics, modalities for selectively modulating the majority of important disease and disorder-relevant alternative splicing targets are currently lacking. Surmounting this obstacle will require new strategies for the efficient and sensitive screening of small molecules to identify leads for further drug development.

Previously described methods for the high-throughput detection of alternative splicing events that are suited to screening small molecule modulators include multiplexed RT-PCR-based assays coupled to detection using electrophoresis or sequencing-based readouts[10–13]. However, these methods are challenging to scale up when requiring the screening of chemical libraries of more than a few thousand compounds. In contrast, single target splicing event reporter assays that couple fluorescence or luminescence-based detection can readily be performed on larger compound libraries, as they are more amenable to robotic handling and plate-based detection and less expensive to scale. Splicing reporters designed for the detection of alternative splicing modulators often employ dual luminescence or fluorescence expression cassettes. In a typical format, a target alternative exon is engineered with a frameshift mutation such that its splicing results in expression of one downstream luminescent or fluorescent protein, whereas its skipping shifts the reading frame to express another luminescent or fluorescent protein with a distinct emission wavelength[14]. Dual fluorescence or luminescence reporter assays have been applied in screens designed to detect new trans-acting splicing regulators in cell lines and during animal development, as well as in small molecule screens[15–21] (reviewed in ref. [22]).

In this study, we describe a dual luciferase splicing reporter system coupled to a high-throughput screening pipeline designed for the sensitive and specific detection of modulators of alternative splicing. We demonstrate the utility of this platform in a screen of ~95,000 small molecules—comprising ~90,000 drug-like compounds and ~5000 bioactive drugs—for modulating the splicing levels of a neuronal microexon (21 nt) in the myocyte-specific enhancer factor 2d (Mef2d) gene, a transcription factor with critical functions in nervous system development. The Mef2d microexon is representative of a broader programme of ~300 human neuronal microexons with multiple functional and genetic links to nervous system development and disorders[23,24]. In particular, this microexon, which functions in transcriptional activation, is highly conserved in vertebrates, regulated by changes in neuronal activity, frequently partially skipped in individuals with autism spectrum disorder (ASD), and controlled by trans-acting regulators that have genetic and functional links to ASD, including the 'master activator' of microexons, the Ser/Arg-repeat matrix protein 4 (SRRM4; also known as nSR100), and the neural enriched splicing regulator RBFOX1[23,25–27]. SRRM4 and its target microexon programme are additionally over-expressed and proposed to act as drivers in a subset of aggressive neuroendocrine cancers[28,29]. Our screen results, together with secondary screens and follow-up mechanistic investigation, identify both small molecule activators and repressors of neuronal microexon splicing. We additionally demonstrate rescue of splicing of microexons within the cerebral cortex of an Srrm4-haploinsufficient mouse model[27] that has multiple ASD-like features. Collectively, the results provide a sensitive and efficient pipeline for conducting small molecule screens for disease and disorder-relevant alternative splicing events, and further establish a resource of compounds that may serve as a basis for the future development of candidate small molecule therapeutics requiring modulation of microexon splicing.

## Results

### Dual Nano and Firefly luciferase reporters for the highly sensitive detection of alternative splicing modulators

To generate reporters with enhanced detection sensitivity and specificity for splicing modulators, we employed a dual detector cassette comprising open reading frames for Nano and Firefly luciferases (abbreviated below as NLuc and FLuc, respectively), cloned downstream of target microexons of interest. Nano luciferase was chosen for its smaller size, reduced stability and approximate 150-fold increase in luminescence signal compared to other fluorescence proteins, whereas Firefly luciferase was chosen for its relatively large degree of emission wavelength separation from NLuc as compared to other widely used luciferases, such as Renilla[30]. An additional advantage of employing NLuc is that its substrate displays relatively enhanced stability and lower background activity. A single nucleotide was inserted within the 21 nt Mef2d microexon (Fig. S1A), such that inclusion or skipping results in a frameshift and expression of FLuc and NLuc, respectively (V1 reporter, Fig. 1A, left-hand panel).

False positives can arise in screens employing dual fluorescence or luciferase reporters when a modulator impacts the expression of a fluorescent or luciferase reporter protein independently of splicing changes[16,31]. Accordingly, a second, reciprocal version of the dual-Luc splicing reporter was generated by deleting a single nucleotide from the Mef2d microexon (Fig. S1A), such that its inclusion and skipping results in expression of NLuc and FLuc, respectively (V2 reporter, Fig. 1A, right-hand panel). True-positive splicing modulators are thus expected to produce comparable, reciprocal changes in the ratios of expression of NLuc and FLuc from both V1 and V2 reporters.

In general, the sensitivity of luciferase-based assays requires modulator-dependent changes in luminescence ratios that are independent of any pre-existing, residual levels of luciferase protein, to ensure that the ratio of luciferase proteins accurately reflects changes to the ratio of alternatively spliced mRNAs. One approach to improve reporter sensitivity is to render luciferase proteins unstable by fusion with a peptide sequence that induces proteolytic degradation[32]. Accordingly, to further enhance the sensitivity of the splicing reporters, we fused the C-termini of the NLuc and FLuc cassettes with PEST degradation sequences (Fig. 1A).

To assess the responsiveness of the V1 and V2 Mef2d microexon reporters, we generated mouse Neuro-2a (N2A) rtTA Flp-In cell lines stably expressing each reporter from the Rosa26 locus, and then monitored changes in the ratios of Firefly and Nano luciferase luminescence in response to increased levels of SRRM4, expressed from a transiently transfected vector (Fig. 1B and C). To afford direct monitoring NLuc and FLuc expression at the protein level, the corresponding ORFs in each reporter were fused N-terminally to three Flag-epitopes, permitting detection with anti-Flag antibody (Fig. 1A). Initially, we observed that the luminescence ratios emitted from cell lines expressing the Mef2d splicing reporter from constitutively active promoters (CMV, PGK or EFS) were minimally responsive to increasing concentrations of SRRM4, likely due to high levels of expression of luciferase prior to induction of SRRM4 expression (data not shown). However, when expressed from a doxycycline (dox)-inducible, minimal CMV promoter following transfection of the SRRM4 expression vector, consistent concentration-dependent responses to SRRM4 were observed for the V1 and V2 reporters (Fig. 1B, C). Further supporting that the detected changes in luminescence ratios are due to alternative splicing, we confirmed reciprocal changes in the ratios of protein isoforms expressed from these reporters in response to expression of SRRM4, by western blotting with anti-Flag antibody (Fig. S1B). A second set of dual-luciferase splicing reporters containing a 9nt microexon from the Shank2 gene, with addition and deletion of single frameshifting nucleotides (Fig. S2A), were generated (Figs. 2A and S2B) and observed to produce comparable results as the Mef2d dual-luciferase

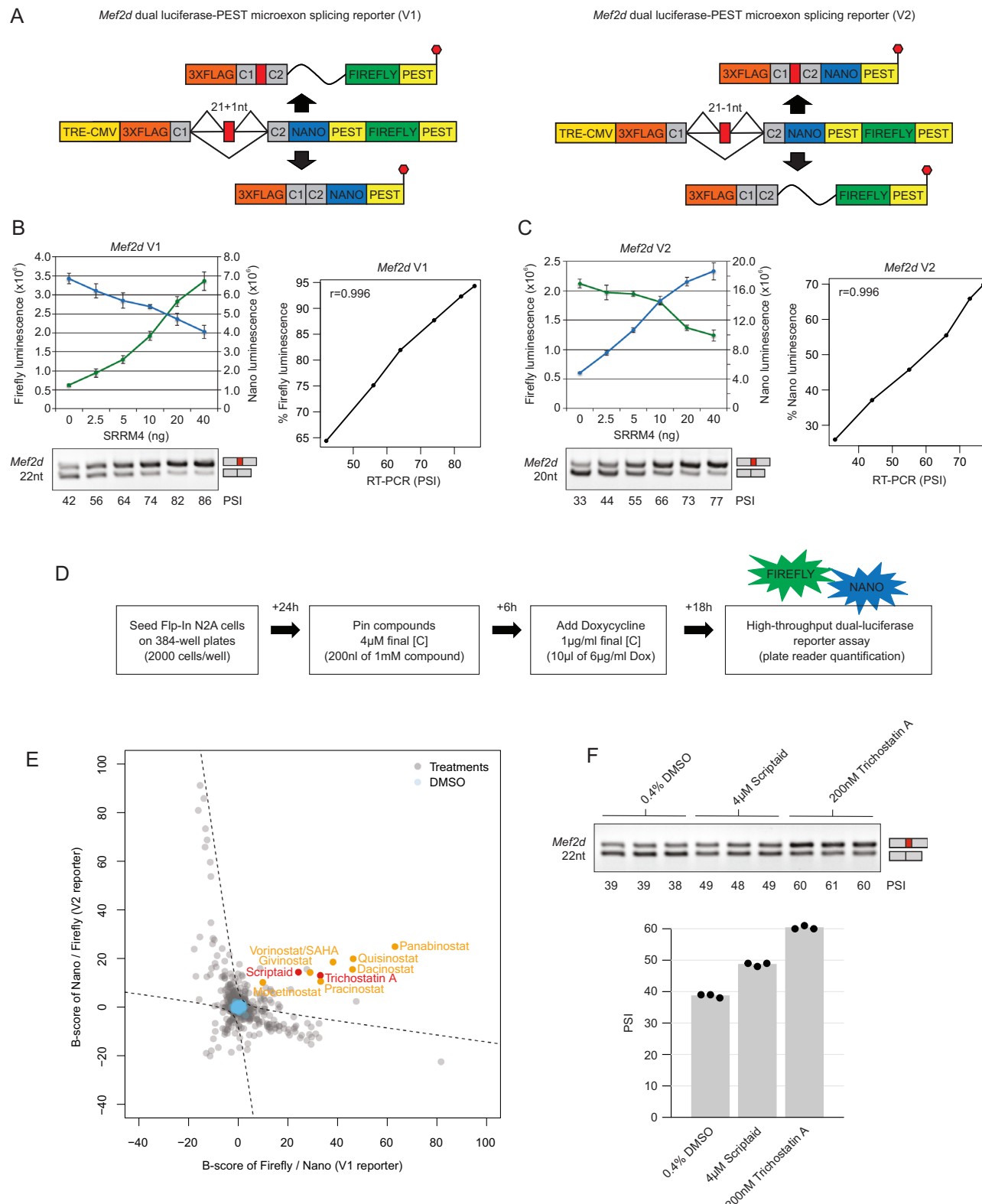

splicing reporters when tested for responses to increasing concentration of SRRM4 (Fig. S2C and S2D).

To assess the linearity of the SRRM4 dose-response of luminescence ratios from the V1 and V2 NLuc and FLuc reporters, and to further confirm that these measurements reflect alternative splicing changes, we performed RT-PCR assays to directly monitor percent spliced-in (PSI) levels of microexon inclusion. This revealed a high correlation between percent Firefly luminescence from the Mef2d V1 reporter, or percent Nano luminescence from the Mef2d V2 reporter, and the corresponding RT-PCR PSI measurements (Pearson correlation = 0.996 for both reporters; Fig. 1B and C, respectively). Comparable results were obtained for the Shank2 V1 and V2 microexon

**Fig. 1 | Generation of Neuro-2A cell lines expressing doxycycline-inducible, dual-luciferase microexon splicing reporters. A** Schematic diagram of reciprocal, dual luciferase microexon splicing reporters expressed in Neuro-2A (N2A) cells under doxycycline-inducible control. The reporters contain a microexon from the Mef2d gene, engineered to introduce a +1nt (V1) or −1nt (V2) shift in the downstream reading frame (see Fig. S1 for details), such that microexon inclusion or skipping results in expression of Nano or Firefly luciferases tagged with PEST protein degradation sequences. The reporters additionally contain 5′ and 3′ flanking native intron and constitutive exon sequences (C1 and C2) from the Mef2d gene, and coding sequence for three tandem Flag epitopes (3 X FLAG) to facilitate detection of expressed luciferases at the protein level (Fig. S1). **B** Top: Quantification of Nano and Firefly luminescence from dual luciferase microexon splicing reporter V1 in response to transfection with increasing amounts of an SRRM4 expression vector (*n* = 4, data are represented as mean values +/− SD). Bottom: RT-PCR assays of reporter Mef2d microexon percent spliced-in (i.e. the percentage of transcripts with the exon spliced in, PSI) in response to transfection with increasing

amounts of SRRM4 expression vector. Right: Correlation between percent Firefly luminescence and ΔPSI from RT-PCR assays (*r* = 0.996). **C** Quantification of Nano and Firefly luminescence from dual-luciferase microexon splicing reporter V2 (*n* = 4, data are represented as mean values +/− SD) as in (**B**) but with the correlation analysis comparing percent Nano luminescence and ΔPSI from RT-PCR assays (*r* = 0.996). **D** Schematic diagram of high-throughput screening protocol using the V1 and V2 Mef2d microexon dual-luciferase splicing reporter cell lines in 384-well format. **E** B-scores of ratios of Firefly and Nano luminescence for the reciprocal Mef2d reporters (V1 and V2) in a pilot screen testing manually curated 'toolkit' and Tocris libraries. HDAC inhibitors are highlighted in orange and red. The B-scores are plotted using the following formulae: Mef2d V1 reporter: (B-score of Firefly V1/Nano V1); Mef2d V2 reporter: (B-score of Nano V2/ B-score of Firefly V2). Hyperbolae were fit to establish hit selection thresholds (see 'Methods' for details). **F** RT-PCR analysis of reporter Mef2d microexon ΔPSI in response to 24 h treatments with 200 nM Trichostatin A (TSA), 4 μM Scriptaid, or DMSO control. Three replicates are shown for each treatment condition.

reporters (Pearson correlation = 0.925 and 0.933, respectively; Fig. S2C and S2D). The Shank2 reporter was found to produce an additional, in-frame spliced product from an alternative 3′ splice site upstream of the Shank2 microexon (Fig. S2E), which may contribute to the reduced linearity of the Shank2 reporter relative to the Mef2d reporter. Notably, for both Mef2d reporters, the insertion or deletion of a single nucleotide to generate the V1 and V2 configurations resulted in a consistent shift in PSI over the range of SRRM4 expression levels. Nonetheless, both configurations displayed highly sensitive and linear responses to increasing concentrations of SRRM4. Comparing the Mef2d and Shank2 V1 and V2 reporters, the baseline (i.e. no induction of SRRM4) PSI of the Mef2d reporters is closer to the mid-range of 50 PSI (i.e. 42 PSI and 33 PSI, respectively) which renders it preferable for screening both activating and inhibiting small molecule modulators. Accordingly, we next employed the Mef2d V1 and V2 NLuc/FLuc dual reporters engineered with PEST sequences to perform a high-throughput screen for small molecule modulators of microexons.

### A high-throughput small molecule screen pipeline for modulators of microexon splicing

We incorporated the reciprocal V1 and V2 Mef2d microexon splicing reporter cell lines in a protocol amenable to robotically manipulated, high-throughput screening of small molecules in 384-well plates (Fig. 1D; 'Methods'). In brief, cells are plated on Day 1, small molecules are dispensed followed by doxycycline-induced expression of the reporter on Day 2, and luminescence measurements are performed on Day 3. To test the performance of this protocol, we initially conducted a small-scale pilot screen using a focused 'toolkit' library consisting of 160 bioactive compounds, together with a Tocriscreen library comprising an additional 1280 small molecules (Fig. 1E). The luminescence ratios of the reciprocal splicing reporters were plotted using 'B-scores'[33], based on the following formulae: V1 reporter: (B-score of Firefly V1/Nano V1); V2 reporter: (B-score of Nano V2/Firefly V2). This revealed false positives, detected as unidirectional luminescence signals from either NLuc or FLuc (Fig. 1E). Therefore, to avoid false positives during hit selection, we generated custom hyperbolae to exclude the majority of compounds that induced unidirectional changes in luminescence (see 'Methods' for details). Importantly, a group of the small molecules resulted in reciprocal changes in the V1 and V2 reporter luminescence ratios and therefore were considered true positives. Interestingly, these included several broad-specificity histone deacetylase (HDAC) inhibitors (Fig. 1E, orange and red labels), including Trichostatin A (TSA) (PubChem CID: 444732) and Scriptaid (PubChem CID: 5186), the effects of which were independently validated by RT-PCR assays (Fig. 1F). Accordingly, TSA and Scriptaid were used as positive control modulators in a larger-scale screen described next.

### A 95,000 small molecule screen reveals positive and negative modulators of microexon splicing

To systematically identify modulators of microexon splicing, we assembled a diverse collection of approximately 95,000 small molecules comprising libraries that include approximately 12,000 drug-like small molecules from the Chembridge Diversity Set, 52,000 drug-like small molecules from the Maybridge HitDiscover collection, 24,000 small molecules from a 'PAIN-free' collection curated from Enamine, ChemDiv and ChemBridge libraries by the Ontario Institute for Cancer Research (OICR), as well as ~5000 bioactive molecules assembled from several smaller libraries, including Tocris, LOPAC, Prestwick and a protein kinase inhibitor (PKI) library.

A primary screen of the small molecule libraries described above was performed using treatment for 24 h at a final concentration of 4 μM for each compound, except for compounds from the PKI library, which were screened at a final concentration of 0.5 μM. The activities of the reciprocal splicing reporters were again plotted using a ratio of B-scores as described above. The positive control compound TSA was assayed at 40 nM, 150 nM or 200 nM final concentrations, and Scriptaid was assayed at 4 μM (Fig. 2A and Supplementary Data 1). As in the pilot screen, hyperbolae were fit to optimize separation of positive and negative controls while excluding the majority of false positives generated from unidirectional changes in luminescence (see 'Methods' for details). Using these hyperbolae to define thresholds, we identified 738 (0.8%) small molecules as putative activators, and 644 (0.7%) small molecules as putative inhibitors (Fig. 2A).

As a secondary round of validation designed to narrow down potential lead compounds and determine their optimal concentrations for further evaluation, we cherry-picked 567 putative hits (Supplementary Data 2) from the primary screen that represent a diverse range of compound classes and performed serial dilutions to assay their effects across ten concentrations (40 nM–20 μM final concentration range, or 13 nM–6.7 μM final concentration range for the PKI library) using the Mef2d V1 microexon reporter only. Figure 2B shows example results corresponding to serial dilutions for the microexon activator HDAC inhibitor compound, SAHA/Vorinostat, and the microexon inhibitor compound AW00693. Notably, as expected AW00693 treatment resulted in an increased expression of Nano luminescence and corresponding decreased expression of Firefly luminescence (Fig. 2B, upper panel), whereas treatment with SAHA/Vorinostat led to an increase in Firefly luminescence but not a reciprocal decrease in Nano luminescence, possibly due to this compound promoting both increased splicing and expression or stability of the reporter transcripts (Fig. 2B, lower panel). The serial dilution assays confirmed similar results as observed for the primary screen for 156 of the 567 compounds (i.e. comparable changes in the NLuc/FLuc luminescence ratios) for a minimum of two

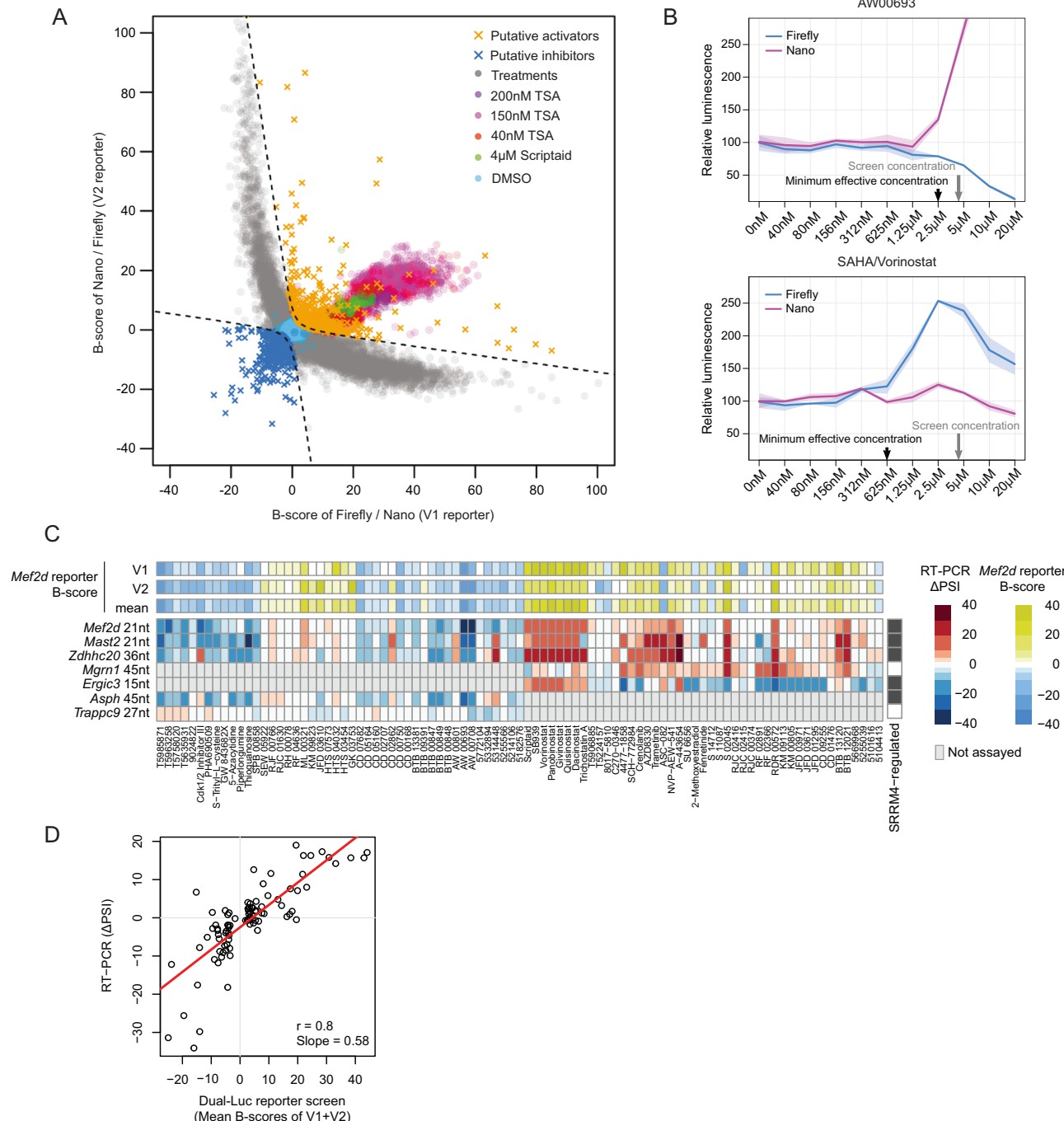

**Fig. 2 | Identification of small molecule modulators of microexon splicing from a high-throughput screen of ~ 95,000 small molecules. A** Identification of small molecule modulators of microexon splicing from a screen of ~95,000 compounds using the V1 and V2 Mef2d microexon reporters shown in Fig. 1A. The B-scores of luminescence ratios for the reciprocal splicing reporters are plotted as described for Fig. 1E. Hyperbolae were fit to establish hit selection thresholds (see 'Methods' for details). **B** A two-fold serial dilution series (40 nM–20 μM) was performed for 567 putative hit compounds selected from (**A**) and assayed using the Mef2d V1 reporter (see 'Methods' for details); example plots are shown for the change in relative luminescence of Nano and Firefly luciferase over the dilution series for the Mef2d microexon repressor AW00693 (top), and the activator SAHA/Vorinostat (bottom).

Shaded area and solid line represent range and mean of two biological replicates. **C** Lower rows: Heatmap showing changes in splicing levels (ΔPSI) of the Mef2d reporter microexon, endogenous microexons (Mef2d, Mast2, Ergic3, Trappc9), and longer cassette neural-differential exons (Zdhhc20, Mgrn1, Asph), following treatment of N2A cells with 91 hit compounds validated by the secondary, serial dilution screen. Upper rows: Luminescence ratio B-scores, based on measurements from the primary screen using the V1 and V2 reporters (**A**), are shown for comparison. RT-PCR data were collected from single replicate experiments. **D** Correlation analysis of ΔPSI measured by RT-PCR with mean changes in luminescence ratio B-scores measured using the primary screen V1 and V2 reporter data in (**A**).

consecutive concentrations. The minimal and maximal effective concentration range for each validated hit was then determined based on reporter activity from the two-fold serial dilutions (Fig. 2B).

## Identification of small molecule inhibitors and activators of neuronal microexon splicing

To further confirm the effects of 91 small molecules on splicing of the Mef2d microexon and assess their impact on the splicing of

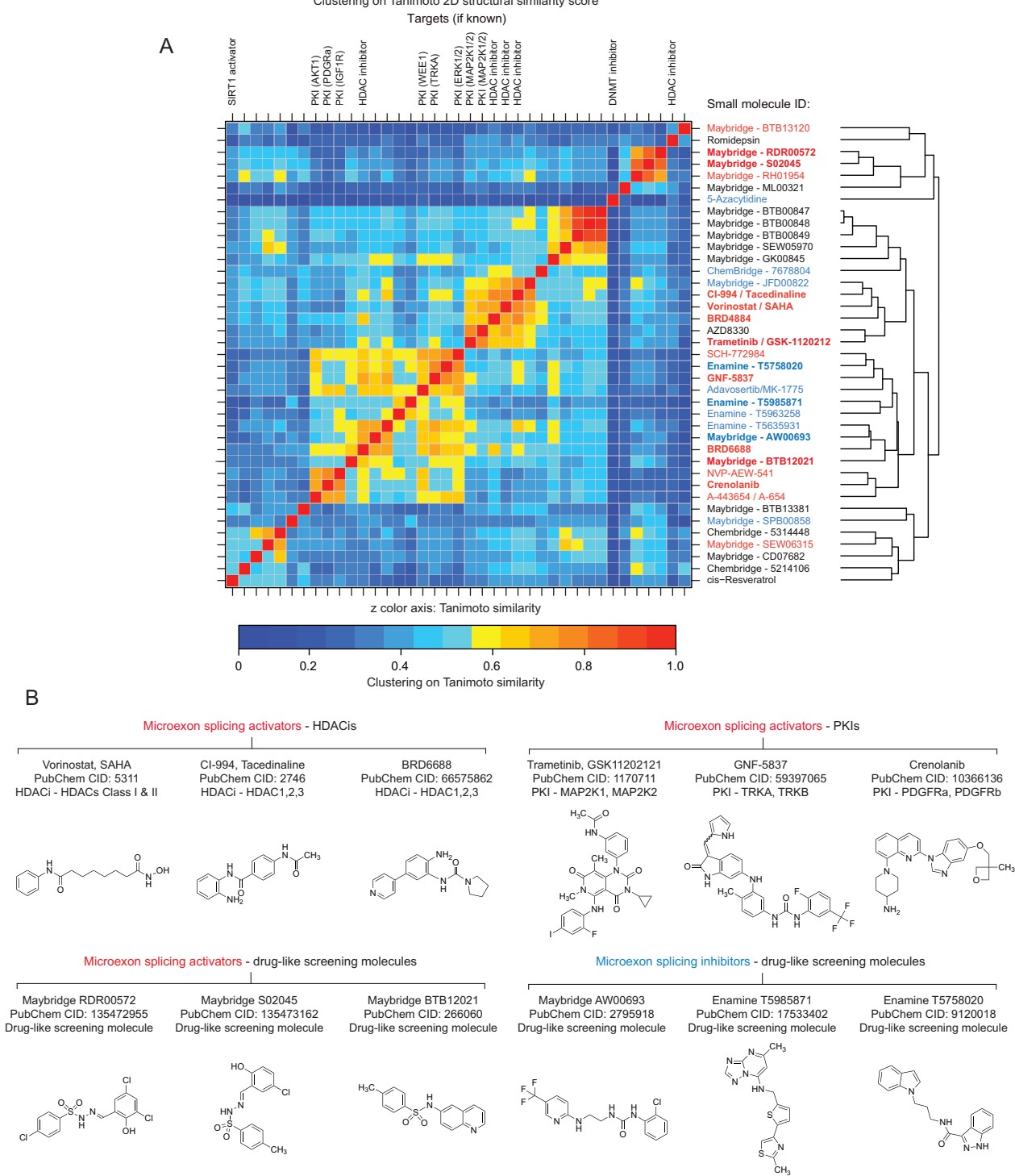

**Fig. 3 | Classification and chemical structures of small molecule modulators of microexon splicing. A** Clustering of representative RT-PCR-validated hits using Tanimoto 2D structural similarity score. Right labels: Representative microexon activator and repressor compounds selected for RNA-Seq analysis are indicated in red and blue text, respectively. Top labels: Histone deacetylase (HDAC) inhibitors and protein kinase inhibitors (PKIs) are indicated along with targets if known. **B** Representative 2D chemical structures representing four categories of small molecule modulators of microexon splicing (HDAC inhibitors, PKIs, and drug-like screening molecule activators and inhibitors). Compounds shown in (**B**) are highlighted in bold in (**A**).

endogenous neural-differential microexons, we treated N2A cells with an optimized concentration (as determined from the two-fold serial dilution series) and performed RT-PCR assays using primer pairs directed to flanking constitutive exon sequences (Fig. 2C and Supplementary Data 3). For small molecules predicted to promote

splicing, we assayed three Srrm4-dependent neural-differential exons in transcripts encoding Mast2, Zdhhc20 and Ergic3 proteins, and for comparison, an Srrm4-independent neural differential exon in Mgrn1 transcripts (Fig. 2C). For small molecules predicted to inhibit microexon splicing, we assayed splicing of exons in Mast2 and Zdhhc20

transcripts, an Srrm4-dependent exon in Asph transcripts and an Srrm4-independent exon in Trappc9 transcripts (Fig. 2C). We additionally performed RT-PCR assays of splicing of the Mef2d reporter microexon (using reporter-specific flanking exon primers). The alternative splicing events selected for RT-PCR validation include both microexons (≤27 nt) (i.e. Trappc9, Ergic3 and Mast2) and longer (>27 nt) neural-regulated exons (i.e. Asph, Mrgn1, and Zdhhc20). The Ergic3 microexon was additionally selected for validation of putative activators since it has relatively low inclusion levels in N2A cells under control conditions. Conversely, the Trappc9 microexon was selected for validation of putative inhibitors since it has relatively high inclusion in N2A cells under control conditions. This selection of test exons thus afforded an initial assessment of potential selectivity of the small molecule effects on microexons versus longer neural exons, as well as for their impact on microexons regulated through Srrm4-dependent versus -independent mechanisms.

Overall, we detected a strong correlation between the changes in FLuc/NLuc ratios from the primary screen and changes in PSI (ΔPSI) measured from the RT-PCR assays of Mef2d microexon inclusion levels from the V1 and V2 reporters (Fig. 2D, mean Pearson's $r = 0.80$). Moreover, most compounds resulted in splicing level changes of endogenous microexons in the same direction as the Mef2d reporter exon; however, several activator compounds reduced inclusion levels of the Ergic3 microexon, presumably reflecting that different microexons have distinct factor requirements and mechanisms of splicing (Fig. 2C).

Following RT-PCR validation, a Tanimoto similarity index was used to cluster representative compounds with activity in microexon activation and repression, based on their 2D structural similarity (Fig. 3A). Among the highest-ranking (i.e. displaying the largest mean, small molecule-induced ΔPSI for Srrm4-dependent microexons, Supplementary Data 3) activators are several broad-spectrum (Class I and II-selective) hydroxamate-based HDAC inhibitors, including Vorinostat/SAHA (PubChem CID: 5311), as well as the HDAC1,2,3-selective ortho-Aminoanilide-based HDAC inhibitors Tacedinaline/CI-994 (PubChem CID: 2746) and kinetically-selective HDAC2 inhibitor BRD6688 (PubChem CID: 66575862) (Fig. 3B, upper left panel). These findings confirm and extend results observed from the pilot screen, demonstrating that multiple HDAC inhibitors that target class 1–3 HDACs promote the splicing of Srrm4-dependent microexons. A second major class of high-ranking activators of microexon splicing is a group of PKIs targeting several components of the MAPK pathway, including GNF-5837 (target: TRKA/B) (PubChem CID: 59397065), Crenolanib (target: PDGFRa/b) (PubChem CID: 10366136), and Trametinib (target: MAP2K1/2) (PubChem CID: 11707110) (Fig. 3B, upper right panel). A significant proportion of the highest-ranked activators also comprise drug-like screening molecules without known targets. The most potent class belongs to a group of benzenesulfonamide-related compounds identified from the Maybridge HitDiscover Collection, including RDR00572 (PubChem CID: 135472955) and BTB12021 (PubChem CID: 266060) (Fig. 3B, lower left panel).

The highest-ranking inhibitors of microexon splicing (i.e. inducing the largest mean ΔPSI for Srrm4-dependent microexons, Supplementary Data 3) are also drug-like molecules without known targets. These compounds originate from several libraries including Enamine and Maybridge and include T5758020 (PubChem CID: 9120018), T5985871 (PubChem CID: 17533402), and AW00693 (PubChem CID: 2795918) (Fig. 3B, lower right panel).

## Transcriptome-wide analysis of small molecule treatments on alternative splicing and gene expression

To globally assess the effect of small molecules on alternative splicing (and gene expression), we selected 27 representative activators and inhibitors and then performed polyA+ RNA-Seq following a 24 h treatment of N2A cells (Fig. 4A). The analyzed compounds were selected for analysis based on: (1) having amongst the largest effect-sizes on endogenous microexon splicing—i.e. largest mean ΔPSI for Srrm4-dependent microexons (Supplementary Data 4), and (2) structural diversity—i.e. compounds that represent distinct structural classes based on clustering of Tanimoto coefficients for 2D structural similarity (Fig. 3A). As a control at the timepoint of RNA isolation, we monitored cell morphology via brightfield microscopy (Fig. S3A), viability via propidium iodide staining (Fig. S3B), and proliferation rate using 5-Ethynyl-2'-deoxyuridine incorporation (Fig. S3C). At 24 h, the compound treatments had a minimal impact on cell morphology and viability of N2A cells (Fig. S3A and S3B), while several compounds, including CI-994, Trametinib, AW00693 and T5985871, reduced cell proliferation (Fig. S3C). The RNA-seq data were analyzed to assess small molecule treatment-induced alternative splicing changes involving microexons (MIC), longer cassette alternative exons (CEs), alternative 5' and 3' splice site selection events (Alt5 and Alt3), and intron retention events (IR) (ΔPSI ≥ 10) (Fig. 4B). The activator and inhibitor compounds assessed by RNA-seq showed varying degrees of selectivity for promoting and repressing the splicing of microexons, as determined by comparing the proportions of total microexons affected by the treatments, relative to the proportions of other alternative splicing events classes affected by the same treatments.

Among the small molecules that display the strongest enrichment for activation (ΔPSI ≥ 10) of the splicing of neural microexons relative to other classes of alternative splicing events is the Class 1–3-specific HDAC inhibitor CI-994. This compound activated 46 of 585 detected microexons (PubChem CID: 2746), 11 of which are 'neural-high' (i.e. highly included in neural cells and tissues). Furthermore, the benzenesulphonamide-related screening compound RDR00572 (PubChem CID: 135472955) activated 48 of 531 microexons (among which 17 are neural-high), and the MEK inhibitor Trametinib (PubChem CID: 11707110) activated 49 of 531 microexons (among which 20 are neural-high) (Fig. 4B, left panel). Moreover, these compounds promoted splicing of 27/116 (23%), 32/115 (28%) and 32/117 (27%) of detected Srrm3/4-regulated microexons[19], respectively (Fig. 4C).

Examples of small molecules that display an enrichment for repressive (ΔPSI ≥ 10) effects on microexons relative to other classes of profiled alternative splicing events include AW00693 (115 of 531 microexons) (PubChem CID: 2795918) and T5985871 (113 of 531 microexons) (PubChem CID: 16352234) (Fig. 4B, right panel). Remarkably, T5985871 and AW00693 repressed the splicing of 79/105 (75%) and 87/115 (76%) of all detected Srrm3/4-regulated microexons, respectively (Fig. 4C). A more comprehensive analysis of alternative splicing changes for all 27 small molecule treatments assessed by RNA-seq is provided in Fig. S4.

We performed a Gene Ontology (GO) analysis for genes containing microexons and longer (>27 nt) cassette exons with changes in splicing inclusion following each small molecule treatment. Consistent with an enrichment for promoting inclusion of Srrm4-dependent microexons, which are concentrated in genes that function in nervous system development and biology[23,24], enriched GO terms included those associated with neural biology for treatments with the HDAC inhibitor CI-994 (i.e. 'neuron projection morphogenesis'), Trametinib ('synaptic membrane') and the drug-like screening molecule RDR00572 ('synaptic vesicle', 'axon part' and 'synapse part') (Fig. 4D). No significantly enriched GO-terms were found associated with splicing changes elicited by the two inhibitor compounds.

We also assessed global changes in gene expression for each of the 27 small molecule treatments. As expected, due to their known impact on transcription, we observed that HDAC inhibitor treatments induce the broadest changes in gene expression. For example, treatment with the HDAC class 1–3 inhibitor CI-994 (PubChem CID: 2746) resulted in 983 up-regulated and 2729 down-regulated genes (FDR < 0.05, LFC > 1, max RPKM > 5), corresponding to 9.6% and 26.7% of 10,289 genes expressed at > 5 RPKM in either compound- or DMSO-

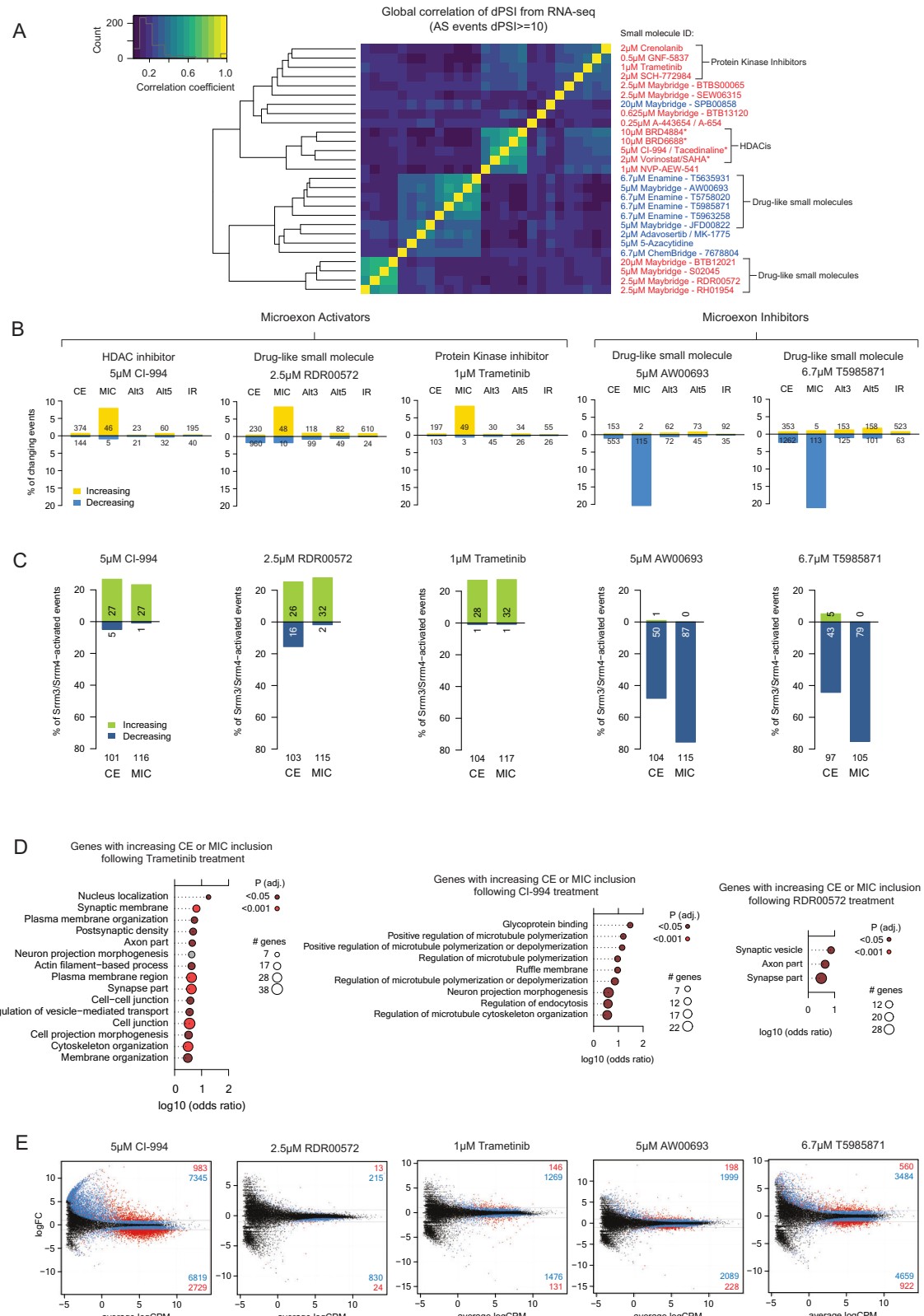

treated cells (Fig. 4E). Notably, two CI-994 derivatives and kinetically selective HDAC2 inhibitors, BRD4884 (PubChem CID: 71465631) and BRD6688 (PubChem CID: 66575862) induced smaller numbers of changes in gene expression than CI-994, while promoting splicing inclusion of a comparable number of microexons (Fig. S5A and S5B). Specifically, these compounds resulted in 570 (5.7%) up-regulated and 454 (4.5%) down-regulated genes, and 498 (5.0%) up-regulated and 622

down-regulated (6.2%) genes, respectively. Notably, we did not observe a significant overlap in gene expression changes for those genes containing microexons with altered splicing levels following HDAC inhibitor treatment (Fig. S5C) (see 'Discussion').

By comparison with the results with HDAC inhibitor treatments, substantially fewer changes in gene expression were detected with the MEK inhibitor Trametinib (PubChem CID: 11707110) (146 up-regulated

**Fig. 4 | Transcriptome-wide analysis of alternative splicing and gene expression changes following small molecule treatments in N2A cells. A** Global correlation of RNA-seq-profiled ΔPSI patterns of alternative splicing events (representing all classes) in N2A cells, following treatments with 27 different small molecules highlighted in Fig. 3 that activate (red) or inhibit (blue) microexon splicing. *indicates analysis of two replicate treatments (otherwise analysis of single treatments was performed). **B** Percentage of detected alternative splicing events belonging to different classes with changing levels (≥10 ΔPSI) following treatments with CI-994, RDR00572, Trametinib, AW00693 or T5985871. CE cassette exons; MIC microexons; Alt5/Alt3 alternative 5′/3′-splice sites; IR intron retention. Absolute numbers of increasing and decreasing events are indicated. See Fig. S4 for analysis of all compounds analyzed by RNA-seq. **C** Percentage of Srrm4/Srrm3-dependent

alternative splicing events with changing levels (≥10 ΔPSI) following treatments with CI-994, RDR00572, Trametinib, AW00693 or T5985871. Absolute numbers of exons with increased and decreased levels are indicated. CE cassette exons; MIC microexons. See Fig. S5B for additional HDAC inhibitors. **D** Gene Ontology (GO) enrichment analysis for genes containing cassette exons (CE) or microexons (MIC) with increased levels (≥10 ΔPSI) following treatment with CI-994, Trametinib or RDR00572. *P*-value thresholds from one-tailed, adjusted *t*-tests as implemented in FuncAssociate are indicated ('Methods'). **E** Global analysis of gene expression (GE) changes (log fold-change [FC]) following treatments with CI-994, RDR00572, Trametinib, AW00693 or T5985871. FDR < 0.05 (blue), FDR < 0.05, LFC > 1, maxRPKM >5 (red)); numbers at the top and bottom right corner indicate genes with increased and decreased expression, respectively.

and 131 down-regulated genes, or 1.5% and 1.4%), and minimal gene expression changes were detected for the benzenesulphonamide-related compound RDR00572 (PubChem CID: 135472955) (13 up-regulated and 24 down-regulated, or 0.1% and 0.3% genes, respectively) (Fig. 4E, left panel). Treatments with the two microexon splicing inhibitor compounds AW00693 (PubChem CID: 2795918) and T5985871 (PubChem CID: 16352234) resulted in 198 (2.0%) up-regulated and 228 (2.3%) down-regulated genes, respectively, and 560 (5.7%) up-regulated and 922 (9.4%) down-regulated genes, respectively (Fig. 4E, right panel).

## Characterization of microexon inhibitor compounds in neuroendocrine cancer cell lines

To further characterize the effects of the microexon inhibitor compounds in a disease-relevant context, we next performed RT-PCR assays to monitor the splicing of Srrm4-dependent microexons in the neuroendocrine small cell lung cancer (SCLC) cell line NCI-82, which aberrantly express SRRM4 and SRRM3, and in which microexons activated by these factors are included. An RNA sample from an untreated human neuroblastoma cell line IMR-32 was also included as a positive control for detection of microexon inclusion. Following a 48 h treatment with four different microexon inhibitor compounds, we observed a substantial reduction in splicing inclusion of all analyzed Srrm4-dependent microexons, as well as a reduction in splicing inclusion of an SRRM4-dependent 50nt cassette exon within REST transcripts, the inclusion of which generates a neural-specific isoform termed REST4 (Fig. 5A). REST is a major transcriptional repressor of neurogenesis genes that is expressed in neural precursor cells and non-neuronal cell types, whereas the REST4 isoform lacks repressive activity and enables expression of neuronal-specific genes, including Srrm4 itself[34].

Similarly, using RT-qPCR assays, we determined the effect of the microexon inhibitor treatments on the expression of a range of neuroendocrine biomarkers, relative to a 0.4% DMSO control treatment (Fig. 5B). Notably, a 5 μM treatment with AW00693 led to a 75% reduction in expression of the key neuroendocrine marker Synaptophysin (SYP), as well as a substantial reduction in several other neuroendocrine markers including REST4, CHGA, CHGB, SRRM3 and SRRM4. We also detected reduced expression of SYP, SRRM3 and SRRM4 in three additional neuroendocrine prostate cancer cell lines; 22Rv1, VCaP and NCI-H660, following similar treatments with AW00693 and T5985871 (Fig. 5C). Taken together, these results confirm the activity of the tested inhibitor compounds in reducing microexon splicing levels, and they further reveal that the effects of AW00693 and T5985871 may, at least in part, be due to these treatments reducing levels of expression of Srrm4 and Srrm3 in human neuroendocrine cell lines that express these factors.

## Mechanism of action of small molecule microexon modulators

To further investigate possible mechanisms by which the analyzed small molecules affect microexon inclusion levels, we used the N2A RNA-Seq data described above (Fig. 4) to systematically assess the

effects of the compound treatments on the expression of all RNA-binding proteins annotated with the GO terms 'mRNA binding' or 'mRNA processing' (Fig. 6A). Confirming results from the RT-qPCR analysis of the neuroendocrine cell lines shown in Fig. 5B and C, RNA-Seq analysis of the treatment with the microexon splicing inhibitor T5985871 revealed significant decreases in Srrm4 and Srrm3 mRNA expression. Moreover, genes encoding additional RBPs known to regulate neural-differential alternative splicing, including Rbfox1, Elval1 and Elavl4, also showed reduced levels of mRNA expression following T5985871 treatment. Conversely, both AW00693 and T5985871 treatments led to a significant increase in mRNA expression of a recently defined negative regulator of microexon splicing, Rbm38[35]. Furthermore, treatment of N2A cells with AW00693 and T5985871 followed by western blotting revealed substantial downregulation of SRRM4, and treatment with AW00693 (but surprisingly not with T5985871) resulted in a significant up-regulation of RBM38 protein levels (Fig. 6B). Consistent with these results, we detect a significant overlap between microexons with decreased inclusion upon treatment with AW00693 or T5985871, and microexons that show increased inclusion upon Rbm38 knockdown (Fig. 6C). Taken together, these results suggest that the inhibitor compounds AW00693 and T5985871 may repress splicing of neuronal microexons through a combination of upstream effects that are manifest through the repression of SRRM4 and SRRM3 expression, and promotion of RBM38 expression.

Interestingly, our western blot analyses further revealed that treatment with the PKI Trametinib led to a significant up-regulation of SRRM4 protein levels and down-regulation of RBM38 protein levels, despite the mRNA transcript levels of these factors not showing significant changes following treatments (Fig. 6A and B). Moreover, an analysis of Srrm3 and Srrm4 iCLIP mapping data[19,36] revealed that exons with increased inclusion upon Trametinib treatment are associated with significant enrichment of Srrm4 and Srrm3 iCLIP reads mapping directly upstream of affected exons (Fig. 6D), coincident with the location of functional Srrm4 binding sites[19]. Taken together, these data suggest that Trametinib leads to an up-regulation of SRRM4, which binds upstream and promotes the splicing of exons affected by treatment with this compound.

To further explore the role of Srrm4 in mediating the effects of the activator compounds, we next asked whether they promote microexon inclusion when Srrm4 is depleted. To assess this, we assayed the impact of the compounds on the splicing of microexons (with varying degrees of dependence on Srrm4) in N2A cells, with and without stable expression of an shRNA that knocks down Srrm4[31] (Fig. S6A). If a compound acts through a mechanism that is independent of any effect on Srrm4 expression, we would anticipate its effect on splicing not to be affected by Srrm4 depletion. Consistent with the data described above suggesting that Trametinib activates microexon splicing in part by promoting Srrm4 protein expression, shRNA depletion of Srrm4 markedly reduced the positive effect of Trametinib on splicing inclusion of the Lphn2

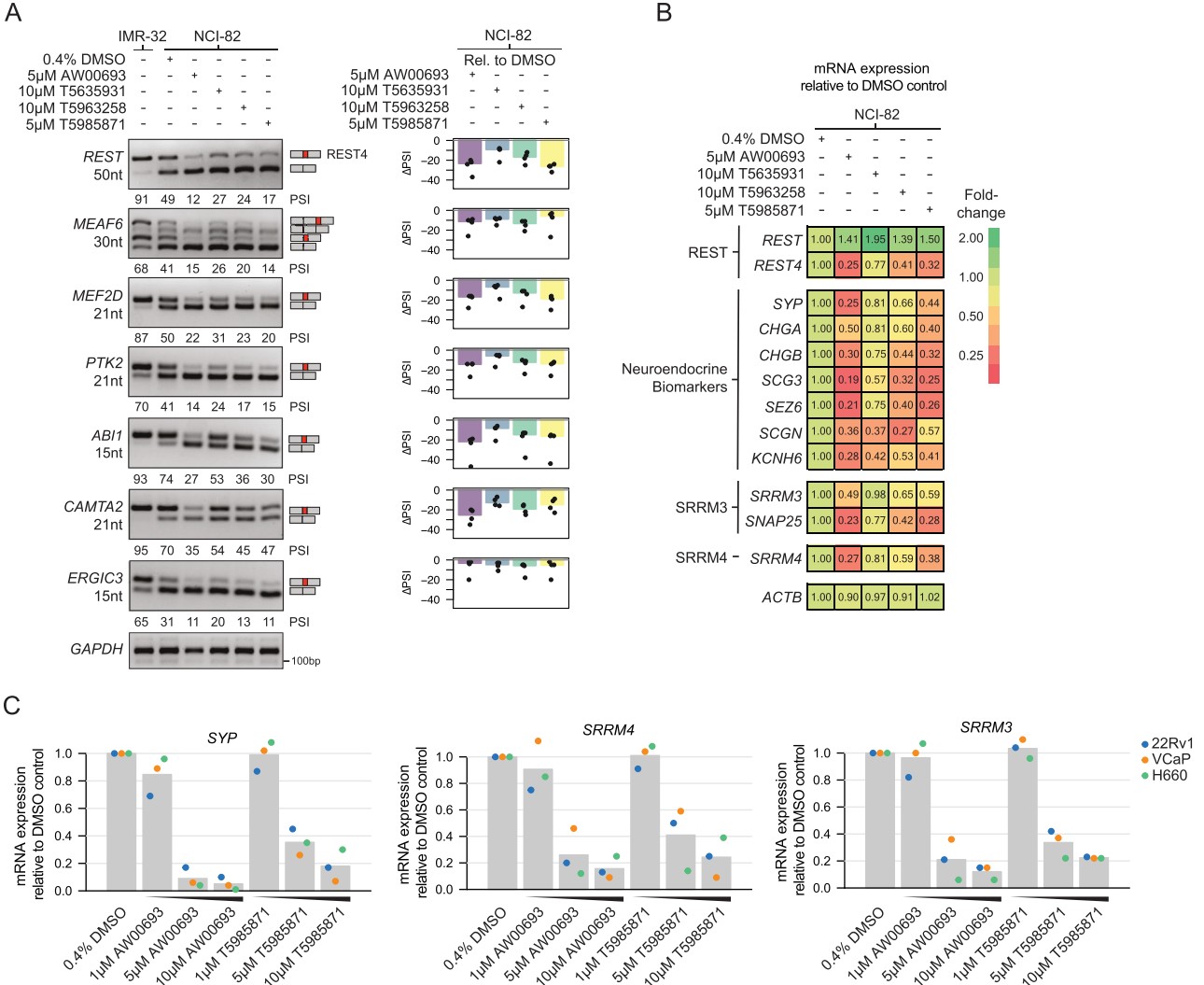

**Fig. 5 | Drug-like screening molecules inhibit microexon splicing and neuroendocrine biomarker expression in neuroendocrine small cell lung cancer (SCLC) and prostate cancer (NEPC)-derived cell lines. A** RT-PCR validation of REST exon 4 (REST4) and microexon splicing changes in NCI-82 cells treated with the drug-like screening molecule inhibitors AW00693, T5635931, T5963258 and T5985871. RNA from an untreated human neuroblastoma cell line IMR-32 was assayed in parallel as a positive control for detection of microexon splicing. Quantification of ΔPSI, relative to 0.4% DMSO treatment, from four replicates is shown on the right. **B** RT-qPCR analysis of mRNA expression level changes for the neuroendocrine cell biomarkers REST/REST4, SRRM3, and SRRM4 in NCI-82 cells

following treatment with the drug-like screening molecule inhibitors AW00693, T5635931, T5963258 and T5985871, relative to 0.4% DMSO control treatment ($n = 3$). **C** RT-qPCR analysis of mRNA expression level changes for the neuroendocrine cell biomarkers SYP, SRRM4 and SRRM3 in the prostate cancer cell lines 22Rv1 (blue), VCaP (orange), and NCI-H660 (green), following single treatments with increasing concentrations (1 μM–10 μM) of the drug-like screening molecule inhibitors AW00693 and T5985871. Expression levels are shown relative to 0.4% DMSO control treatment. Mean mRNA expression levels across all three cell lines are indicated by heights of grey bar plots.

and Zmynd8 microexons. Conversely, the microexon activation effects of the HDAC inhibitors CI-994 and BRD6688 were only modestly reduced in cells depleted of Srrm4, suggesting that these compounds largely do not act through effects on Srrm4 expression. Similarly, treatment with the drug-like screening molecule RDR00572 led to substantial increases in microexon splicing inclusion, in particular for the Zmynd8 and Sorbs1 microexons, both in mock-depleted and Srrm4-depleted N2A cells, suggesting that this compound likely functions through an Srrm4-independent mechanism. Interestingly, combining RDR00572 treatment with the HDAC inhibitor BRD6688 led to additive increases in splicing levels for all tested microexons (Fig. S6B). Taken together, these data suggest that Trametinib and the HDAC inhibitors BRD6688 and CI-994 promote microexon splicing at least in part by promoting Srrm4 expression, whereas the drug-like molecule RDR00572 appears to act independently of Srrm4.

## Microexon rescue in an ASD mouse model

Finally, to assess the potential in vivo utility of the screen-defined small molecule microexon activators, we investigated their activity in rescuing microexon splicing in the cerebral cortex of a mouse model of ASD that is haploinsufficient for Srrm4 ($Srrm4^{+/\Delta7-8}$)[27]. Previously, we demonstrated that this mouse model displays broad disruption of the Srrm4-dependent neuronal microexon splicing programme and multiple neuroanatomical and behavioural ASD-like features[27]. Given previous reports[37,38] that administering HDAC inhibitors to murine models of neurodevelopmental disorders can, in some cases, reverse ASD-like phenotypes, we tested the HDAC inhibitor SAHA/Vorinostat in our mouse model. For comparison, we also tested the microexon activator RDR00572. To assess efficacy, we performed single daily subcutaneous injections of these compounds between postnatal day 2 (P2) and postnatal day 8 (P8) (Fig. 7A). At P8, total RNA was isolated from the cerebral cortex and

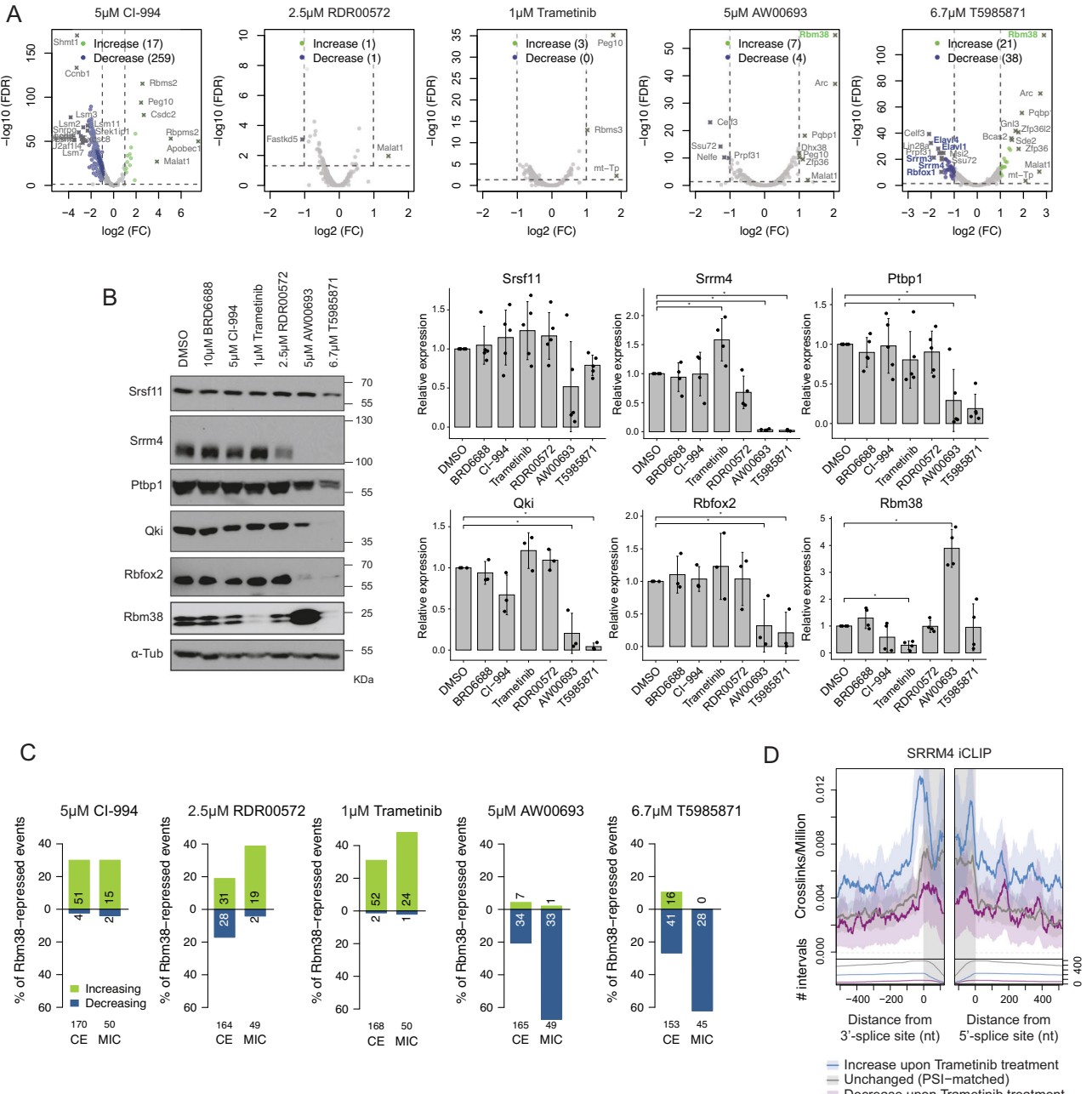

**Fig. 6 | Srrm4-dependence for small molecule modulation of microexon splicing. A** RNA-Seq analysis of mRNA expression level changes corresponding to mRNA-binding proteins, following treatments with compounds CI-994, RDR00572, Trametinib, AW00693 or T5985871, using data shown in Fig. 4. Levels are shown relative to 0.4% DMSO control treatments. Genes changing more than 2-fold and with FDR < 0.05 are highlighted, with thresholds indicated with dashed lines. **B** Western blot analysis of changes in levels of splicing regulators in N2A cells following treatment with compounds BRD6688, CI-994, RDR00572, Trametinib, AW00693, or T5985871, in comparison with levels detected in 0.4% DMSO control treatment. *$p < 0.05$, one-sided paired samples student's $t$ test (with three or more independent biological replicates). Data are represented as mean +/− SD. $p$ values: $p = 0.025$ (SRRM4, Trametinib); $3.1 * 10^{-7}$ (SRRM4, AW00693); $2.7 * 10^{-7}$ (SRRM4, T5985871); 0.0077 (PTBP1, AW00693); 0.00028 (PTBP1, T5985871); 0.015 (QKI,

AW00693); 0.00034 (QKI, T5985871); 0.050 (RBFOX2, AW00693); 0.024 (RBFOX2, T5985871); 0.0015 (RBM38, Trametinib); 0.0019 (RBM38, AW00693). **C** Percentage of cassette exons (CE) and microexons (MIC) repressed by Rbm38 that change in level (≥10 ΔPSI) following treatment with CI-994, RDR00572, Trametinib, AW00693 or T5985871. Absolute numbers of exons with increased and decreased levels are indicated. **D** Meta-exon plot of individual-nucleotide resolution UV crosslinking and immunoprecipitation (iCLIP) signal (measured as reads recovered from crosslinked RNA per million reads) of Srrm4 surrounding cassette exons and microexons in cortical neurons, and corresponding to exons that change in splicing level following Trametinib treatment of N2A cells. Lines and shaded areas indicate mean and standard error across the sets of events and three replicate experiments. Lower plots indicate number of aligned sequences at each position.

four Srrm4-dependent microexons, selected on the basis of displaying Srrm4 dependency in vivo[27] and responding to treatment with SAHA/Vorinostat or RDR00572 in N2A cells, were analyzed by RT-PCR assays (Fig. 7B). We did not observe rescue of microexon splicing in *Srrm4$^{+/\Delta7-8}$* mice following treatment with SAHA/

Vorinostat (see 'Discussion'). However, treatment with RDR00572 resulted in rescue of splicing to levels that are comparable to those detected in the cerebral cortex of wild-type mice for all four analyzed Srrm4-dependent microexons (Fig. 7B). In contrast, the splicing levels of several microexons that did not display Srrm4-

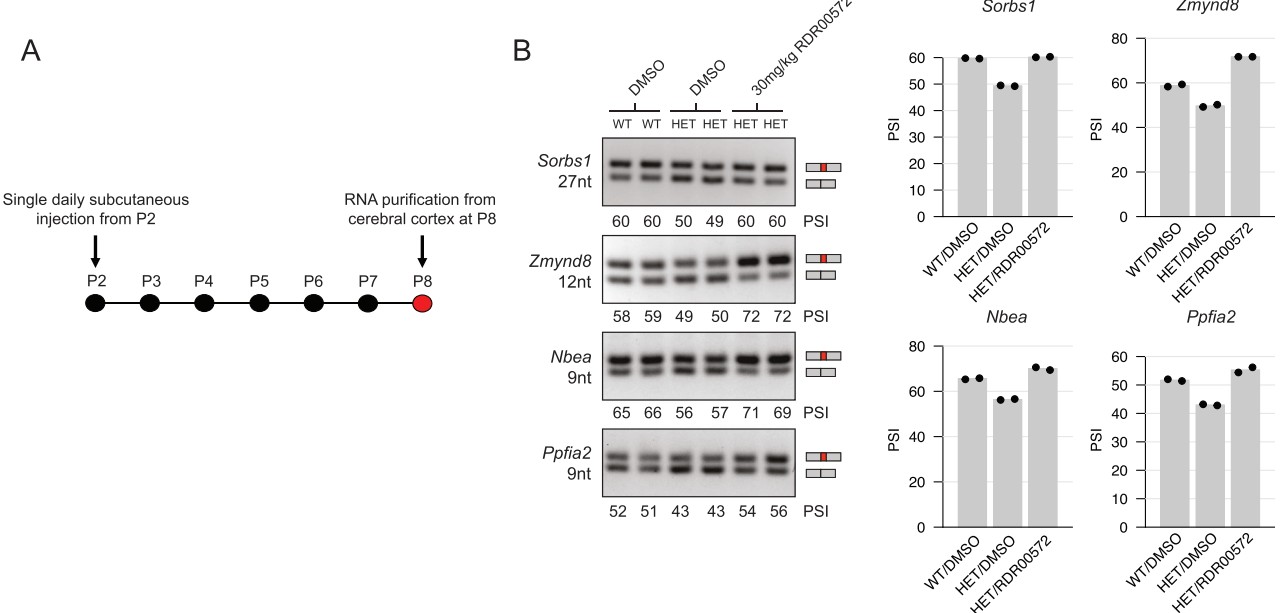

**Fig. 7 | Rescue of splicing of several analyzed brain microexons following treatment of mice haploinsufficient for Srrm4 with the orphan compound RDR00572. A** Protocol for postnatal treatment in which pups received a single daily subcutaneous injection between postnatal day 2 (P2) to postnatal day 8 (P8). RNA was purified from the cerebral cortex of treated mice at P8. **B** Left, RT-PCR assays monitoring levels of splicing of Srrm4-dependent microexons in mock-treated (DMSO) wild-type (*SRRM4*[+/+]) mice, heterozygous (*SRRM4*[+/Δ7−8]) mice[27], and in *SRRM4*[+/Δ7−8] mice treated with 30 mg/kg doses of RDR00572 (*n* = 2). Right, quantification of replicate experiments.

dependence in vivo were not affected by either compound treatment (data not shown).

## Discussion

In this study, we describe a dual luciferase splicing reporter and its application in a workflow optimized for the facile screening of tens of thousands of small molecules for modulatory effects on alternative exons of interest. The combination of features incorporated in the reporter, including the application of Nano and Firefly expression cassettes and fusion of PEST peptides directing luciferase turnover, as well as features of the screening protocol itself, collectively provide a sensitive, quantitative and high-throughput approach for screening splicing modulators. We demonstrate the utility of our system in a screen directed at the identification of modulators of neuronal microexons. This screen successfully identified small molecules that positively or negatively alter the splicing levels of the screen-assayed Mef2d microexon, as well as of a broader programme of neural microexons that is disrupted in ASD and activated in neuroendocrine cancers.

The small molecule screen identified several compounds with known and unknown targets that, relative to other classes of alternative splicing events, preferentially promote or repress microexons. These include HDAC inhibitors, PKIs, and drug-like benzenesulphonamide-related compounds. Notably, several Class I and II-selective HDAC inhibitors promote splicing of a substantial fraction (25–30%) of detected neural microexons. This observation is intriguing in light of the growing number of studies employing HDAC inhibitor administration to enhance performance in memory and learning tests and in the (partial) reversal of phenotypes associated with different neurological disorders and diseases including autism, epilepsy, deafness and SMA[37–43]. These wide-ranging effects of HDAC inhibitors are consistent with their dose-dependent, broad impact on gene expression changes that encompass activation of disease and disorder-related genes and pathways that may be reduced due to genetic haploinsufficiency, or else distinct disease-related mechanisms that affect chromatin state[43]. In this regard, a previous study provided evidence that HDAC1/2, in addition to regulating the expression of ASD-related genes in the embryonic mouse brain, also forms interactions with multiple ASD-related proteins in human neuronal cells that display preferential expression during foetal brain development, thereby potentially impacting ASD-related processes at multiple levels[44].

Recently, it was reported that low concentrations of HDAC inhibitor treatment can alleviate an RNA polymerase II block caused by the ASO Nusinersen used to treat SMA patients[43]. The ASO was found to promote deposition of the histone silencing mark H3K9me2 on the SMN2 gene, creating a roadblock to RNA polymerase II elongation that acts negatively on the splicing of SMN2 exon 7, reducing its functional activity (refer to Introduction). HDAC inhibition counteracted this effect of the ASO to result in increased exon 7 inclusion without significant effects on overall SMN2 transcript levels. In analogous manner, we observed an additive effect when combining the HDAC inhibitor BRD6688 with the drug-like orphan compound RDR00572 in promoting the splicing of microexons. However, at the concentrations of HDAC inhibitors and other small molecules used in the present study to modulate microexons, and based on our downstream mechanistic analyses, it is likely that many of the observed splicing changes are predominantly due to indirect yet selective effects on the expression of specific microexon regulators, such as Srrm4 and Rbm38. Lastly, while we were unable to rescue microexon splicing in vivo with the HDAC inhibitor SAHA/Vorinostat, possibly due to the limited capacity of this compound to cross the blood-brain barrier, administering mice with the drug-like screening molecule RDR00572 efficiently rescued the splicing of tested Srrm4-dependent microexons in the cerebral cortex of Srrm4-haploinsufficient mice. Future studies will be directed

at establishing the extent to which this compound, as well as its derivatives, can rescue microexon deficiency in vivo.

In conclusion, our study collectively provides a powerful screening approach for modulators of splicing as well as a resource of small molecules that serves as a basis for future investigations of the regulatory pathways that impact microexon splicing, and the development of potential in vivo applications. We envision the application of our high-throughput screening platform extending to additional approaches for modulating splicing in future studies, including testing ASOs and CRISPR-based methods. Indeed, in parallel work, the Mef2d microexon splicing reporter described in the present study was used in a screen of deactivated (d)CasRx-splicing factors that led to the discovery of an efficient and specific guide RNA-directed activator of target alternative exons[45]. Further studies will be necessary to further determine the precise mechanism of action and potential of the small molecule microexon modulators identified in the present study in clinical applications.

## Methods

The research in this study was conducted in compliance with the Animals for Research Act of Ontario and the Guidelines of the Canadian Council on Animal Care. The Centre for Phenogenomics (TCP) Animal Care Committee reviewed and approved all procedures conducted on animals at TCP.

### Cell culture

All cell lines used in this study were sourced from ATCC. Mouse neural 2A (N2A) cells were grown in DMEM supplemented with 10% FBS, sodium pyruvate, MEM non-essential amino acids, and penicillin/streptomycin. CGR8 mouse embryonic stem cells (ESCs) were cultured as described previously on gelatin-coated plates (Gabut et al., 2011; Han et al., 2013). Human neuroblast IMR-32 cells and VCaP cells were grown in DMEM supplemented with 10% FBS. NCI-H82 and 22Rv1 cells were grown in RPMI-1640 medium supplemented with 10% FBS. NCI-H660 cells were grown in RPMI-140 medium supplemented with 0.005 mg/mL Insulin, 0.01 mg/mL Transferrin, 30 nM sodium selenite (final concentration), 10 nM Hydrocortisone (final concentration), 10 nM beta-estradiol (final conc.), 4mM L-glutamine (final conc.), and 5% FBS. All cell lines were maintained at 37 °C with 5% $CO_2$.

### Dual Nano and Firefly luciferase reporter generation

**Engineering dual-luciferase microexon splicing reporters.** We generated a Nano luciferase/Firefly luciferase cassette, and a Nano luciferase-PEST/Firefly luciferase-PEST cassette, using overlapping gBlocks gene fragments synthesized by IDT. The Nano luciferase ORF was engineered to remove termination codons from alternative reading frames so that when the Nano luciferase ORF was out of frame, translation of the downstream Firefly luciferase ORF would still occur. The sequences of the gBlocks gene fragments are provided below:

**Nano luciferase/Firefly luciferase fragment 1.** GGTCTTCACACTCGAAGATTTCGTTGGGGACTGGCGACAGACAGCCGGCTACAACCTGGACCAAGTCCTCGAACAGGGAGGTGTGTCCAGTTTGTTTCAGAATCTCGGGGTGTCCGTAACTCCGATCCAAAGGATTGTCCTGAGCGGCGAAAATGGGCTGAAGATCGACATCCATGTCATCATCCCGTACGAAGGTCTGAGCGGCGACCAAATGGGCCAGATCGAAAAAATTTTCAAGGTGGTGTACCCTGTGGACGATCATCACTTCAAGGTGATCCTGCACTATGGCACACTGGTAATCGACGGGGTTACGCCGAACATGATCGACTATTTCGGACGGCCGTACGAAGGCATCGCCGTGTTCGACGGCAAAAAGATCACTGTAACAGGGACCCTGTGGAACGGCAACAAAATTATCGACGAGCGCCTGATCAACCCCGACGGCTCCCTGCTGTTCCGAGTAACCATCAACGGAGTGACCGGCTGGCGGCTGTGCGAACGCATTCTGGCGTAACAGAAGACGCCAAAAACATAAAGAAAGGCCCGGCGCCATTCTATCCGCTGGAAGATGGAACCGCTGGAGAGCAACTGCATAAGGCTATGAAGAGATACGCCCTGGTTCCTGGAACAATTGCTTTTACAGATG

**Nano luciferase/Firefly luciferase fragment 2.** ACGATTTTGTGCCAGAGTCCTTCGATAGGGACAAGACAATTGCACTGATCATGAACTCCTCTGGATCTACTGGTCTGCCTAAAGGTGTCGCTCTGCCTCATAGAACTGCCTGCGTGAGATTCTCGCATGCCAGAGATCCTATTTTTGGCAATCAAATCATTCCGGATACTGCGATTTTAAGTGTTGTTCCATTCCATCACGGTTTTGGAATGTTTACTACACTCGGATATTTGATATGTGGATTTCGAGTCGTCTTAATGTATAGATTTGAAGAAGAGCTGTTTCTGAGGAGCCTTCAGGATTACAAGATTCAAAGTGCGCTGCTGGTGCCAACCCTATTCTCCTTCTTCGCCAAAAGCACTCTGATTGACAAATACGATTTATCTAATTTACACGAAATTGCTTCTGGTGGCGCTCCCCTCTCTAAGGAAGTCGGGGAAGCGGTTGCCAAGAGGTTCCATCTGCCAGGTATCAGGCAAGGATATGGGCTCACTGAGACTACATCAGCTATTCTGATTACACCCGAGGGGGATGATAAACCGGGCGCGGTCGGTAAAGTTGTTCCATTTTTTGAAGCGAAGGTTGTGGATCTGGATACCGGGAAAACGCTGGGCGTTAATCAAAGAGGCGAACTGTGTGTGAGAGGTCCTATGATTATGTCCGGTTATGTAAACAATCCGGAAGCGACCAACGCCTTGATTGACAAGGATGGATGGCTACATTCTGGAGACATAGCTTACTGGGACGAAGACGAACACTTCTTCATCGTTGACCGCCTGAAGTCTCTGATTAAGTACAAAGGCTATCAGGTGGCTCCCGCTGAATTGGAATCCATCTTGCTCCAACACCCCAACATCTTCGACGCAGGTGTCGCAGGTCTTCCCGACGATGACGCCGGTGAACTTCCCGCCGCCGTTGTTGTTTTGGAGCACGGAAAGACGATGACGGAAAAAGAGATCGTGGATTACGTCGCCAGTCAAGTAACAACCGCGAAAAAGTTGCGCGGAGGAGTTGTGTTTGTGGACGAAGTACCGAAAGGTCTTACCGGAAAACTCGACGCAAGAAAAATCAGAGAGATCCTCATAAAGGCCAAGAAGGGCGGAAAGATCGCCGTGTAA

**Nano luciferase-PEST/Firefly luciferase-PEST fragment 1.** GTCTTCACACTCGAAGATTTCGTTGGGGACTGGCGACAGACAGCCGGCTACAACCTGGACCAAGTCCTCGAACAGGGAGGTGTGTCCAGTTTGTTTCAGAATCTCGGGGTGTCCGTAACTCCGATCCAAAGGATTGTCCTGAGCGGCGAAAATGGGCTGAAGATCGACATCCATGTCATCATCCCGTACGAAGGTCTGAGCGGCGACCAAATGGGCCAGATCGAAAAAATTTTCAAGGTGGTGTACCCTGTGGACGATCATCACTTCAAGGTGATCCTGCACTATGGCACACTGGTAATCGACGGGGTTACGCCGAACATGATCGACTATTTCGGACGGCCGTACGAAGGCATCGCCGTGTTCGACGGCAAAAAGATCACTGTAACAGGGACCCTGTGGAACGGCAACAAAATTATCGACGAGCGCCTGATCAACCCCGACGGCTCCCTGCTGTTCCGAGTAACCATCAACGGAGTGACCGGCTGGCGGCTGTGCGAACGCATTCTGGCGAATTCTCACGGCTTTCCGCCCGAGGTCGAAGAGCAAGCCGCCGGTACATTGCCTATGTCCTGCGCACAAGAAAGCGGTATGGACCGGCACCCAGCCGCTTGTGCTTCAGCTCGCATCAACGTCTAACAGAAGACGCCAAAAACATAAAGAAGGCCCGGCGCCATTCTATCCGCTGGAAGATGGAACCGCTGGAGAGCAACTGCATAAGGCTATGAAGAGATACGCCCTGGTTCCTGGAACAATTGCTTTTACAGATGCACATATCGAGGTGGACATCACTTACGCTGAGTACTTCGAAATGTCCGTTCGGTTGGCAGAAGCTATGAAACGATATGGGCTGAATACAAATCACAGAATCGTCGTATGCAGTGAAAACTCTCTTCAATTCTTTATGCCGGTGTTGGGCGCGTTATTTATCGGAGTTGCAGTTGCGCCCGCGAACGACATTTATAATGAACGTGAATTGCTCAACAGTATGGGCATTTCGCAGCCTACCGTGGTGTTCGTTTCCAAAAAGGGGTTGCAAAAAATTTTGAACGTGCAAAAAAAGCTCCCAATCATCCAAAAAATTATTATCATGGATTCTAAAACGGATTACCAGGGATTTCAGTCGA

**Nano luciferase/Firefly luciferase fragment 1.** (continued) CACATATCGAGGTGGACATCACTTACGCTGAGTACTTCGAAATGTCCGTTCGGTTGGCAGAAGCTATGAAACGATATGGGCTGAATACAAATCACAGAATCGTCGTATGCAGTGAAAACTCTCTTCAATTCTTTATGCCGGTGTTGGGCGCGTTATTTATCGGAGTTGCAGTTGCGCCCGCGAACGACATTTATAATGAACGTGAATTGCTCAACAGTATGGGCATTTCGCAGCTACCGTGGTGTTCGTTTCCAAAAAGGGGTTGCAAAAAATTTTGAACGTGCAAAAAAAGCTCCCAATCATCCAAAAAATTATTATCATGGATTCTAAAACGGATTACCAGGGATTTCAGTCGATGTACACGTTCGTCACATCTCATCTACCTCCCGGTTTTAATGAATACGATTTTGTGCCAGAGTCCTTCGATAGGGACAAGACAATTGCACTGATCATGAACTCCTCTGGATCTAC

TGTACACGTTCGTCACATCTCATCTACCTCCCGGTTTTAATGAATAC
GATTTTGTGCCAGAGTCCTTCGATAGGGACAAGACAATTGCACTGAT
CATGAACTCCTCTGGATCTACTGGTC

**Nano luciferase-PEST/Firefly luciferase-PEST fragment 2.** GAATA
CGATTTTGTGCCAGAGTCCTTCGATAGGGACAAGACAATTGCACTG
ATCATGAACTCCTCTGGATCTACTGGTCTGCCTAAAGGTGTCGCTCT
GCCTCATAGAACTGCCTGCGTGAGATTCTCGCATGCCAGAGATC
CTATTTTTGGCAATCAAATCATTCCGGATACTGCGATTTTAAGTGTT
GTTCCATTCCATCACGGTTTTGGAATGTTTACTACACTCGGATATT
TGATATGTGGATTTCGAGTCGTCTTAATGTATAGATTTGAAGAAGAG
CTGTTTCTGAGGAGCCTTCAGGATTACAAGATTCAAAGTGCGCT
GCTGGTGCCAACCCTATTCTCCTTCTTCGCCAAAAGCACTCTGATTG
ACAAATACGATTTATCTAATTTACACGAAATTGCTTCTGGTGGCGCTC
CCCTCTCTAAGGAAGTCGGGGAAGCGGTTGCCAAGAGGTTCCAT
CTGCCAGGTATCAGGCAAGGATATGGGCTCACTGAGACTACATCAG
CTATTCTGATTACACCCGAGGGGGATGATAAACCGGGCGCGGTC
GGTAAAGTTGTTCCATTTTTTGAAGCGAAGGTTGTGGATCTGGATAC
CGGGAAAACGCTGGGCGTTAATCAAAGAGGCGAACTGTGTGTGAGA
GGTCCTATGATTATGTCCGGTTATGTAAACAATCCGGAAGCGACCAA
CGCCTTGATTGACAAGGATGGATGGCTACATTCTGGAGACATAGCT
TACTGGGACGAAGACGAACACTTCTTCATCGTTGACCGCCTGAAG
TCTCTGATTAAGTACAAAGGCTATCAGGTGGCTCCCGCTGAATTGG
AATCCATCTTGCTCCAACACCCCAACATCTTCGACGCAGGTGTCGCA
GGTCTTCCCGACGATGACGCCGGTGAACTTCCCGCCGCCGTTGTTGT
TTTGGAGCACGGAAAGACGATGACGGAAAAAGAGATCGTGGAT
TACGTCGCCAGTCAAGTAACAACCGCGAAAAAGTTGCGCGGA
GGAGTTGTGTTTGTGGACGAAGTACCGAAAGGTCTTACCGGAAA
ACTCGACGCAAGAAAAATCAGAGAGATCCTCATAAAGGCCAAGAAG
GGCGGAAAGATCGCCGTGAATTCTCACGGCTTTCCGCCCGAGGTCG
AAGAGCAAGCCGCCGGTACATTGCCTATGTCCTGCGCACAAGAAAGC
GGTATGGACCGGCACCCAGCCGCTTGTGCTTCAGCTCGCATCAACG
TCTAA

We next utilized sequences described previously (Gonatopoulos-Pournatzis et al., 2018) in which conserved microexons from the murine Shank2 (ENSE00002478436/ENSMUSE00001006087) and Mef2d (ENSE00001054660/ENSMUSE00000673645) genes were mutated via the insertion or deletion of a single nucleotide, such that microexon inclusion would result in a change to the downstream reading frame of the dual-luciferase or dual-luciferase-PEST cassettes. The sequences contain a mutated microexon, its flanking introns, the upstream constitutive exons and 20–50 nt of the downstream constitutive exon, and were PCR-amplified using a forward primer that included attB1 sites required for cloning products into the pDONR221 entry vector using Gateway recombination. Next, a single Nano luciferase/Firefly luciferase or Nano luciferase-PEST/Firefly luciferase-PEST cassette was created by performing an overlap extension PCR, using the overlapping gBlocks gene fragments, and a reverse primer that included attB2 sites required for cloning products into the pDONR221 entry vector using Gateway recombination. The two PCR products were joined into a single fragment using Gibson Assembly (NEB) via the overlapping ends and the combined fragment was then further amplified using the forward primer of the minigene fragment and the reverse primer of the dual-luciferase cassette. The final PCR products containing both attB1 and attB2 sites were then cloned into the pDONR™221 (entry vector) using Gateway recombination cloning technology. Edited sequences were subcloned into a customized pcDNA5-based Gateway compatible vector (Life Technologies™) that contained 8x TET response elements (TRE) followed by a miniCMV promoter, Kozak sequence, 3xFlag-tag sequence and attR sites. The pcDNA5 vectors containing the dual-luciferase microexon reporters were subsequently used for the generation of N2A Flp-In cell lines.

**Generation of stable N2A Flp-In rtTA3 cell lines.** Doxycycline inducible N2A Flp-In rtTA3 stable cell lines were generated as previously described (Gonatopoulos-Pournatzis et al., 2018). Briefly, N2A Flp-In rtTA3 stable cell lines capable of inducible cDNA expression were created through co-transfection of 200 ng of a rtTA3 compatible pcDNA5/FRT plasmid (modified from Thermo Fisher Scientific, V6010-20), with 2 µg of pOG44 Flp-recombinase expression vector (Thermo Fisher Scientific, V600520). Cell lines with successful cDNA integration were selected and maintained using 200 mg/mL Hygromycin.

**High-throughput small molecule screening**
**Primary screen.** Screening was performed at the Lunenfeld-Tanebaum Research Institute (LTRI) Network Biology Collaborative Centre. The primary screen was performed using reciprocal versions of the Mef2d dual-luciferasePEST splicing reporters. A diverse collection of ~95,000 small molecules, consisting of ~90,000 synthetic drug-like screening compounds and ~5000 bioactive drugs were assembled for high-throughput screening. The libraries included ~12,000 drug-like small molecules from the Chembridge Diversity Set, ~52,000 drug-like small molecules from the Maybridge HitDiscover collection, ~24,000 small molecules from a 'PAIN-free' collection curated from Enamine, ChemDiv and ChemBridge by the Ontario Institute for Cancer Research (OICR), as well as approximately ~6000 bioactive molecules assembled from several smaller libraries, including TOCRIS, LOPAC, Prestwick and a PKI library.

Doxycycline (Dox) inducible N2A Flp-In rtTA3 stable cell lines were seeded at 2000 cells per well in 50 µL medium in 384-well plates (Biomek FX Laboratory Automation Workstation, Beckman Coulter). 200 nL of compounds and DMSO controls were added using a dedicated pin tool (Multimek, Beckman) to achieve final concentrations of 4 µM (except for the PKI library, which was screened at a final concentration of 0.5 µM), for in total 24 h treatment. Six hours post compound treatments, 10 µL of 6 µg/mL Dox was added (1 µg/mL final concentration) to induce the expression of dual-luciferasePEST splicing reporters. Luminescence was detected using the Nano-Glo Dual-Luciferase Reporter Assay System (Promega, N1610) and quantified using a plate reader (Envision, Perkin Elmer).

**Small molecule screening analysis**
**Calculation of B-scores.** Following luminescence measurements, B-scores[41] were calculated independently for each 384-well plate as follows. The ratios between absolute luminescence of the wells in a plate were calculated as Firefly V1/Nano V1 for reporter V1, and Nano V2/Firefly F2 for V2. Luminescence ratios were transformed by iteratively removing column, plate, and row medians until further changes converged below a minimum residual value using a slightly modified R function medpolish from the 'stats' R package, adapted for masking out controls. The ratio of the residual value per well to the median of all absolute plate residuals was used as B-score and was used for downstream analysis and hit selection (see below and Supplementary Data 1).

**Hit selection criterion.** Putative hits were selected by plotting the B-scores of ratios for Firefly and Nano luminescence for each reciprocal reporter (Fig. 2A). Hyperbolae were fitted to optimally separate positive from negative controls while excluding the majority of false positives representing unidirectional changes in luminescence. Custom hyperbolae were generated according to the following formula:

$$(x - y/-7) * (-y + x/-7)/-2 = 4$$

where $x$ and $y$ are B-score of (Firefly V1/Nano V1) and B-score of (Nano V2/Firefly V2), respectively. Using these hit selection criteria, we identified 738 (0.8%) small molecules as putative activators, and 644 (0.7%) small molecules as putative inhibitors.

The assay, library, screen and post-screen analysis are summarized in Supplementary Table 1.

**Secondary serial dilution screen**. As a secondary round of validation, we selected 567 putative hits (Supplementary Data 2) for cherry-picking and performed serial dilutions to assay their effect on splicing reporter luminescence across 10 concentrations (40 nM–20 μM final concentration range), using the Mef2d dual-luciferase-PEST splicing reporter V1. The screening including cell seeding, small molecule treatments, and luminescence reading was performed same as the primary screen. The minimal and maximal effective concentration range for each of the small molecules from the secondary screen was then defined based on reporter activity from the 2-fold serial dilutions.

### RNA extraction, RT-PCR, and RT-(q)PCR assays

Total RNA was extracted using the QIAGEN RNeasy Mini Kit. RT-PCR assays were performed using the QIAGEN OneStep RT-PCR kit. 20 ng of total RNA was used per 10 μl of reaction. The number of amplification cycles was 22 for Gapdh, and 27–32 for all other transcripts analyzed. Reaction products were separated on 1.5–4% agarose gels. Quantification of isoform abundance was performed as previously described (Han et al., 2013). For RT-(q)PCR, first-strand cDNAs were synthesized from 1–2 μg of total RNA using the Thermo Scientific Maxima H Minus First Strand cDNA Synthesis Kit, as per the manufacturer's instructions, and diluted to 20 ng/μl. qPCR reactions were performed in a volume of 10 μl using 1 μl of diluted cDNA and the Roche Applied Science FastStart Universal SYBR Green Master Kit. Primer sequences used for RT-PCR and RT-(q)PCR reactions are available upon request.

### Cell viability, cell proliferation assays and flow cytometry analysis

N2A cells were seeded into wells of 6-well plates 24 h prior to 24 h treatment of small molecule compound. For cell viability assay, cells were harvested by trypsinization, pelleted by centrifugation at $300 \times g$ for 5 min, washed once with DPBS (Gibco), then resuspended in flow cytometry buffer (HBSS/0.1%BSA). Propidium iodide was added to cells 5 min before flow cytometry acquisition, at a concentration of 1 ug/mL.

For cell proliferation assay, cells were treated with 10 μM EdU (5-ethynyl-2′-deoxyuridine) for 60 min prior to being harvested as described above. For each condition, $1 * 10^6$ cells were fixed with 4% PFA for 15 min at room temperature. After a single PBS wash, cells were permeabilized with PBS/0.5% Triton X-100 for 10 min at room temperature, washed once again with PBS, then stained with freshly prepared EdU staining buffer (Click-iT™ EdU Cell Proliferation Kit Alexa fluor 555, Thermo Fisher) for 30 min in dark at room temperature. After stopping the reaction, cells were washed three times with PBS/0.1% Triton X-100, then immediately subjected to flow cytometry acquisition.

For flow cytometry, cells were filtered through a 40 μM mesh filter before flow cytometry. All samples were analyzed using BD LSR Fortessa (BD Biosciences) flow cytometer. Acquired data were analyzed with FlowJo Software (FlowJo, LLC). Single cells were selected using forward scatter and side scatter as standard. Negative gating for staining was determined using PI unstained cells (for cell viability assay) or cells treated with replication inhibitor Aphidicolin (for proliferation analysis, data not shown).

### Immunoblotting and quantification

Cells were lysed in Tris lysis buffer (10 mM Tris-HCl pH7.9, 150 mM NaCl, 1% NP-40, 10% glycerol) and sonicated. Lysate was cleared by centrifugation at 4 °C, $17000 \times g$ for 20 min, supernatants were collected, mixed with Laemmli buffer, and heated 5 min at 95 °C. Samples were separated on variable percentages SDS-PAGE gels and transferred to PVDF membranes. Blots were incubated overnight with the following primary antibodies at the specified dilutions in 5% milk: Mouse anti-a-Tubulin (Sigma-Aldrich T6074) at 1:5000; mouse anti-PTBP1 (Invitrogen 32-4800) at 1:1000; mouse anti-QKI (EMD Millipore MABN624)

at 1:500; rabbit anti-Rbfox2 (Bethyl A300-864A-T) at 1:1000; Rabbit anti-Srsf11 (Thermo Fisher PA5-37056) at 1:1000; Rabbit anti-Srrm4 (Calarco et al., 2009) at 1:5000; Rabbit anti-Rbm38 (Abcam ab200403) at 1:1000.

Band intensities were quantified using ImageJ and the relative expression of proteins was calculated as fold enrichment relative to DMSO control, normalize for a-Tubulin protein level within the same replicates. Paired sample Student's t-test was performed across five biological replicates.

### In vivo rescue experiments

Mice used in this study were C57BL/6. All drug treatments were performed on wild-type and heterozygous mice generated from $nSR100^{+/\Delta7-8}$ inter-crosses and fostered by $nSR100^{+/\Delta7-8}$ females. Beginning at postnatal day 2, wild-type or heterozygous pups received a daily subcutaneous injection consisting of a mock treatment (75% Physiological Saline, 20% Kolliphor EL and 5% DMSO) or a 30 mg/kg dose of RDR00572 (75% Physiological Saline, 20% Kolliphor EL and 5% DMSO/RDR00572). Each pup received seven injections between P2 and P8. At P8, brains were harvested, and the cerebral cortex was surgically dissected and recovered. Total RNA was extracted by resuspending and dissolving the cerebral cortex in RLT buffer supplemented with 1% β-mercaptoethanol, followed by homogenization using Qiagen QIAshredder columns, and purification using QIAGEN RNeasy Mini Kit, following manufacturer's instructions.

### RNA-seq analysis

RNA-seq was performed on two biological replicates from N2A cells treated with the HDACis CI-994, BRD6688, BRD4884 and SAHA/Vorinostat. RNA-seq was performed on single biological samples for the remaining compound treatments.

**AS analysis.** For AS analysis, RNA-Seq data were processed using the vast-tools pipeline, version 2.1 https://github.com/vastgroup/vast-tools[23,46,47]. Changes were considered significant if they were greater than 10 ΔPSI/dPIR and the expected minimum change was different from zero at $p > 0.95$ according to vast-tools' *diff* module. Events were filtered for coverage and junction balance, requiring a vast-tools quality column *score 3* of SOK/OK/LOW for cassette exon (CE), microexon (MIC), and alternative 5′ or 3′ splice site (Alt5/3) events or at least 15 reads for intron retention (IR) events and a balance score (*score 4*) of 'OK' or 'B1' for alternative exons or >0.05 for intron retention events, in at least half of replicate experiments.

**Expression analysis.** GE changes were analyzed by pseudo-aligning pre-trimmed reads to GENCODE vM21 transcripts using Salmon v0.14.1[48] and aggregated per gene using the R package tximport[49] Differential expression was then assessed using the classic mode (exactTest) in edgeR[50], and genes with an FDR < 0.05 and a greater than 2-fold change were considered differential. For each comparison, only genes expressed at a minimum of five RPKM in one or both conditions were considered.

### Gene ontology analyses

GO enrichment analysis for genes containing alternative splicing changes was performed with FuncAssociate 3.0[51] using a custom background of all genes that contained measurable AS events (after filtering as detailed above) of the type in question requiring at minimum a 5-fold enrichment and excluding categories of more than 1000 genes. Terms with a mutual gene overlap of greater then 50% were merged such that only the term with the lowest $p$-value was retained.

### Structural similarity analysis

Tanimoto scores from https://pubchem.ncbi.nlm.nih.gov/score_matrix/score_matrix.cgi. were used, selecting the 2D option.

PubChem compound identifiers (CID) for each of the compounds were obtained from the manufacturer of the chemical library, conversion tools (e.g. http://www.chembridge.com/conversion_tool/index.php), or from researching the Internet on the names or symbols of the compounds. Plotting was performed in R using the levelplot function from the lattice package.

## Reporting summary

Further information on research design is available in the Nature Portfolio Reporting Summary linked to this article.

## Data availability

The data supporting the findings of this study are available from the corresponding authors upon request. Raw and/or normalized luminescence data from the primary and secondary screens are available in Supplementary Data 1. RNA-Seq data generated in the course of this study are available on GEO as series GSE228599. Source data are provided with this paper.

## Code availability

Custom R code used to normalize data and score hits from luminescence reporter screens is available upon request. R scripts used to normalize RNA-Seq data for differential expression and splicing analysis are available on GitHub at https://github.com/UBrau/ProcessDE (v0.5.1); Ref: https://doi.org/10.5281/zenodo.10265546 and https://github.com/UBrau/ProcessVast (v0.5.0); Ref: https://doi.org/10.5281/zenodo.10407129, respectively.

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

## Acknowledgements

The authors thank Thomas Gonatopoulos-Pournatzis, Jonathan Roth and Jack D. Li for helpful discussions and technical assistance, and the Donnelly Sequencing Centre is gratefully acknowledged for sequencing samples. This research was funded by grants from the Simons Foundation Autism Research Initiative Pilot Grant (#513581) (B.J.B), Canadian Institutes of Health Research Foundation Grant (FDN #353947) (B.J.B), and Canada First Research Excellence Fund Medicine by Design Program (J.L.W. and B.J.B). A.J.B was supported by Postdoctoral Fellowships from EMBO and CIHR. H.H. was supported by the Donnelly Centre Home Fellowship and a MITACS Award. B.J.B holds the University of Toronto Banbury Chair in Medical Research and the Canada Research Chair in RNA Biology and Genomics.

## Author contributions

Project conceptualization: A.J.B., H.H. and B.J.B.; Study design: A.J.B., H.H., U.B. and B.J.B., with input from other authors; Experiments: A.J.B., M.W., H.H., S.F., J.J.L. and L.C.C.; Technical assistance: M.J., J.W. and A.D.; Data analysis: U.B., A.P., H.H. and A.J.B.; Data processing: U.B., A.P. and H.H., Manuscript figure preparation: A.J.B., U.B., M.W. and H.H.; Manuscript writing: B.J.B., A.J.B., H.H. and U.B., with input from other authors; Supervision: B.J.B., S.C., R.A. and J.L.W; Funding acquisition: B.J.B. and J.L.W.

## Competing interests

The authors declare no competing interests.
