## [Peer Review File · Nature Communications]

High-throughput sensitive screening of small molecule modulators of microexon alternative splicing using dual Nano and Firefly luciferase reportersREVIEWER COMMENTS

Reviewer #1 (Remarks to the Author):

The manuscript describes a screen of small molecules modulating the inclusion of microexons. The authors use a dual luciferase reporter system containing a microexon. Depending on its splicing behavior (included or skipped), a different luciferase protein is generated. These reporters are used in two reciprocal configurations to minimize false positives resulting from an effect on the activity of one of the luciferase genes. As the results show, this is highly necessary, as for the majority of tested compounds with an effect on the ratio between the Luciferase readouts this is due to an effect on the activity of one of the Luciferase enzymes rather than the splicing event.

The authors use this reporter system to screen collections of small compounds and identify a set of molecules affecting the splicing behavior of the reporter. In general, with the exception of the positive controls, the effect sizes on the splicing reporter readout tend to be small. However, and somewhat surprisingly, they are detectable in RT-PCR validations, which should be much less sensitive than the Luciferase assay.

In order to assess the specificity of the effect on microexons, the authors tried to quantify the effect of some small molecule hits on splicing and gene expression transcriptome-wide. This provides a rich dataset, but the analysis presented in the paper stays very much at the surface. The RNA-seq data could be analyzed more in depth, which might provide additional insights into the selectivity of the effect and enrich the paper as well as provide evidence for molecular targets (e.g. RBPs beyond Srrm4) of the small molecules. Which features of a splicing event determine whether it is affected by a small molecule? Weak consensus sequences? The presence of specific sequence motifs? The authors provide such an analysis on the level of expression of RBPs, but if the claim is a selective effect on a subset of splicing events (all or some microexons), a splice site-centric analysis could be more convincing. At this point, the large number of RBPs showing changes in their gene expression levels rather sheds some doubt on the potential specificity of the effect to microexons.

In general, the manuscript presents an interesting screening strategy that is – for the most part – carefully controlled and described. The authors avoid overselling their results which makes the manuscript more accessible. However, - and maybe as a consequence of unclear results – the text is sometimes vague and therefore not too informative. While the principle of the approach and the identification of small molecules with a potentially preferential effect on microexons will certainly be of interest for researchers in the field, I am hesitant to endorse publication of the manuscript in its present form in a venue with a broad readership like Nature Communications.

Specific points and suggestions:

Abstract, line 9: In light of the results presented in the paper, the word “selectively” is not accurate. The small molecules identified here do not appear to be selective for regulating microexon inclusion levels but to some extent affect other splicing events as well and expression levels of a substantial number of genes in general.

Figure 1BC: There seems to be a systematic difference in PSI between the two configurations (B and C). The authors should comment on this.

Figure 2A: Putative activators and inhibitors with the strongest effect in one configuration typically do not show any effect in the other configurations. Why are they still considered candidates? In addition, the effect size of many “putative inhibitors” or “activators” is close to 0. Was there no cutoff on effect size? What is considered a significant effect? In addition, the use of semi-quantitative RT-PCR

validations for such small effect sizes is questionable? Validation with qPCR should be provided for at least some of the candidates (across the range of effect sizes that the authors deem relevant).

Figure 2C: Why do both readouts (Firefly and Nano) increase in the left panel (or stay more or less unchanged in the case of Nano) while in the right panel one increases as the other one decreases (which seems to be the expected behavior if I understand the readout correctly)?

Figure 2D: Why were some only tested with activators and others only with repressors? And especially in these cases, the directionality of the effect is often lost (activators having a repressive effect and vice versa), including for microexons. These results are not sufficiently explained and discussed.

Lines 214-217: "This selection of test exons afforded an initial assessment of potential selectivity of the small molecule effects on microexons versus longer neural exons, and for their impact on microexons regulated through Srrm4-dependent versus -independent mechanisms." – This sentence is very vague. In light of the results this is understandable, but the statement could still be rephrased to convey a clearer message.

Line 219: 39 top-ranked compounds: what was the cutoff used? Why the top 39?

Figure 3A by itself is not very informative, maybe highlighting the groups presented in 3B or annotating other relevant clusters could add to the graph.

Figure 4C: There seems to be a problem with the normalization of the RNA-seq counts. CPM was used, so the logFC should still be distributed around 0 (which is not the case for panels 2, 3 and 5).

Figure 5BC: What is the proposed mechanism of action of these drugs? This is not a consequence of aberrant microexon splicing (if anything it might be a cause of the observed effect, given that Srrm4 is one of the targets). How do they affect expression of splicing regulators such as Srrm4?

Reviewer #2 (Remarks to the Author):

In this manuscript, Best AJ et al. screened small molecules to identify compounds that regulate micro exons. They employed dual Nano and Firefly luciferase alternative splicing minigenes, modeling screening reporters previously used by various labs (Hagiwara, Cooper, Black). The authors found that HDAC inhibitors and PKI inhibitors were the top hits. They performed RNA-seq of cell lines treated with several selected drugs to determine the breadth of gene expression and splicing changes. The results indicated that these drugs affected expression of SRRM3/4. The authors knocked down SRRM4 in N2a cells and showed that this abolished the drug effect on a couple of micro exons.

The manuscript's strength lies in the scale of the screening effort, which involved testing ~95k small molecules. However, there is insufficient novelty nor new mechanistic insight in the findings, and the authors need to temper their conclusions. For instance, the HDAC inhibitors are clearly not "selective" activators of micro exons. SRRM3/4 are already well known to control micro exons, and the data supporting their involvement in the HDAC inhibitors' activity is relatively weak.

The authors position the manuscript for translational impacts by highlighting the roles of micro exons in developmental disorders. However, no data implicates the translatability of the identified hits. The analyses stop short at cell lines that are irrelevant to a disease setting. The manuscript also lacks sufficient details and statistical analyses to assess the data quality.

Major points

The authors can enhance the comprehensibility of the screen results for readers. For example, it is unclear why different groups of exons were tested for different drug groups (fig 2d). for benchmarking, the same exon groups should be tested unbiasedly for these selected drug groups. The authors can easily acquire data for every drug-exon combination since the drug-treated samples and the splicing assay are available. This is important to evaluate the consistency of the impact of each of 156 drugs on the micro exons vs longer exons. The authors should present the result in a more informative manner (e.g., a heatmap and others) than a simple violin plot in fig 2d. In addition, for transparency, the authors should release the PSI values of the tested exons upon drug treatment from the RT-PCR experiments, e.g., in a supplementary excel file.

The paper and the screen aim to identify small molecules that affect micro exon splicing, as the title and abstract suggest. However, the paper has yet to demonstrate the specificity of candidate drugs in affecting micro exons vs longer alternative exons. The RNA-seq data actually show they affect both.

Moreover, each HDAC inhibitor affects expression of 10-35% genes and the splicing of 10-20% of micro exons. The authors have not presented convincing evidence to indicate specific effects of HDAC inhibitors on micro exon splicing vs global transcriptomic changes.

The potential novelty of the manuscript is the identification of PKI inhibitors, but the authors have not followed up to provide mechanistic insights.

The authors indicate that the micro exons are important in the brain. Therefore, the positive hits should be tested in the brain or neurons to evaluate their effects on micro exons.

It is necessary to specify in the main text which nucleotide is added to and deleted from the mef2d microexon and at what position. This would help readers to understand the design. An important question is to what extent these modifications affect the basal splicing outcome and the screening results. These should be tested by RT-PCR in comparison to the minigene and discussed in detail, given the authors' attempt to present the values of their methods.

To improve readers' understanding of the minigene system, the authors should explain why it does not respond to SRRM4 when expressed from the CMV or PGK promoter? Additionally, the authors should specify which promoters were used for the single luminescence constructs in fig S1b?

Fig1, it would be helpful for readers to know whether the NLuc sequence included in both the inclusion isoform (V1) and the V2 exclusion isoform. The authors should also present Western blot data to show the 3xflag-tagged firefly protein is expressed as the expected size, to ensure its levels and activities reflect the designated splicing pathway other than some unknown processing of the minigene.

Fig1b-c: since NLuc and FLuc are separate assays producing different output units, their relative comparison based on these arbitrary units can be swung easily in either direction. The authors should explain how they achieved such high linearity and clarify why the other reporters do not appear as linear.

Fig 1e: it is unclear what statistics and cutoffs were used to define positive hits.

Line 222 "Among the highest-ranking activators of microexon splicing are several". Line 239 "The highest-ranking inhibitors of microexon splicing are also drug-like molecules" where can readers find the ranking information of drugs?

Line 247 "To globally assess the effect of small molecules on alternative splicing (and gene expression), we selected 27 top representative..." the selection criteria were mentioned but it is unclear how they were applied. The authors should clarify this information in the manuscript so readers understand if there is any selection bias. Treatment conditions are also unclear.

Fig 5, the authors should include data and images of the cell health and density at the time of rna collection as a quality control for the splicing and gene expression analysis.

Fig 6, protein expression and activity are often uncorrelated with their mRNA expression. The authors should test protein expression of these RBPs.

Fig6, the RNA-seq data do not appear to have biological replicates and statistics?

Fig6c, validation using a few SRRM4 targets is underpowered, especially when these exons are different from those used in other figures. The authors should use RNA-seq to provide unbiased results.

Are other (or most) HDAC inhibitors identified as hits in the screen?

Minor points

Fig S2b-c: what are the * isoforms?

statistics is missing in fig 5a, 6c

Reviewer #3 (Remarks to the Author):

Summary:

The authors establish and demonstrate a reciprocal dual luciferase reporter system for screening libraries of molecules for splice-modifying activities. They use this reporter system to screen a large library of small molecules for compounds that promote or inhibit microexon inclusion. They validate representative hits from this screen using various techniques including RNA-seq, qPCR, minigene assays, and dose-response measurements. Several of the compounds identified as promoting microexon inclusion are histone deacetylase (HDAC) inhibitors. The paper proposes that HDAC inhibitors promote microexon inclusion by up-regulating *Srrm4* and down-regulating *Ptbp2*, and that compounds that inhibit microexon inclusion do so by down-regulating *Srrm4* and *Srrm3*.

There is a great deal of interest in screens for splice-modifying drugs, and I believe the reporter system described here is an important contribution to those screening efforts. In particular, Figure 2B convincingly shows that that, had the authors used a non-reciprocal screening system, they would have identified many false-positive hits (i.e., drugs that affect the activity of one or both luciferase readouts, but do not affect splicing of the reporter transcript).

I believe that the core contribution of this paper—the reciprocal dual reporter system—is thoughtful, rigorous, and has the potential to substantially impact screening efforts to identify splice-modifying drugs. However, the data provided to support the mechanistic hypothesis of how identified compounds function are less convincing. The authors may wish to focus more on the screening aspect of their manuscript, and soften their statements about mechanism.

Major comments:

1. The mechanisms proposed in Figure 6D, and discussed in 370-374, seem overly speculative. The changes in expression observed by RNA-seq are suggestive, but they do not on their own show mechanism. The data in Figure 6C does show that CI-994 activity was abolished by *Srrm4* knockdown, which provides evidence that CI-994 acts through an *Srrm4*-dependent mechanism. But BRD6688 activity was abolished by *Srrm4* knockdown at only 2 of the 4 loci assayed. I also do not see evidence that the HDAC inhibitors are functioning by down-regulating *Ptbp2* expression, or that any of the

microexon inhibitor compounds are functioning by down-regulating Srrm3 or Srrm4 expression. Note that HDAC inhibitors have recently been shown to regulate splicing by reducing H3K9me2 and increasing H3K9ac at the affected exon (PMID: 35688133). Is it possible that the identified HDAC inhibitors are regulating microexons by a similar mechanism?

2. The manuscript lacks clarity on whether biological or technical replicates were performed for some of the experiments. To avoid confusion, this information should be explicitly mentioned in the main text or figure legends. For instance, Figures 5A, 5C, and 6A-C do not contain information about biological or technical replicates. Moreover, uncertainties should be reported for the PSI measurements in Figures 5A and 6C, gene expression measurements in Figures 5C and 6A, and Δ PSI measurements in Figure 6B.

3. Some of the data visualizations might be improved.

i. For Figure 5C, instead of the bar plot and points, it might be better to show separate titration curves for each microexon.

ii. For Figure 6A, volcano plots having specific genes of interested highlighted would better illustrate the fold-change and FDR.

iii. For Figure 6B, the data for each of the four microexons assayed would be better displayed separately, and error bars should be shown for each microexon/treatment combination. Statements about additivity and synergy should be supported by quantitative comparisons of Δ PSI values under different treatments.

4. It would be good to quantify protein levels by western blot to more definitively establish the gene expression changes observed in Figure 6A that are argued to contribute to drug mechanism.

Minor comments:

1. The authors should mention that this screening system can be applied to screens that do not involve small molecules, e.g., to ASO libraries or to CRISPR libraries.

2. It might make sense to cite this study, which has discussed regulation of microexons by proteins involved in chromatin organization: Genome-wide CRISPR-Cas9 Interrogation of Splicing Networks Reveals a Mechanism for Recognition of Autism-Misregulated Neuronal Microexons, PMID: 30388412.

3. Gene names should be italicized.

4. In Figure 2D, it might make sense to group the microexons by size, and note in the plot which are Srrm4-dependent.

5. In Figure 2D, it is unclear why so many RT-PCR validated activators inhibit splicing of the Ergic3 microexon, and why some RT-PCR validated inhibitors activate splicing in the other microexons. Please clarify in the main text.

6. Did the identified drugs affect cell viability? Were there any differences in cell morphology or proliferation rate? One might expect this would happen with small molecule drugs at micromolar concentrations, and it might be relevant for mechanism.

7. In line 321 there is a typo; I suggest the correction "expression the REST4 isoform" -> "expression

of the REST4 isoform”.

8. The phrase “whereas the effects of...” in line 372 is confusing because the microexon inhibitors are also proposed to act in an Srrm4-dependent manner.

9. In supplemental Figure 2, “*” needs to be explained in the figure legend as a nonspecific product. It might make sense to sequence this product and report what it is.

RESPONSE TO REVIEWER COMMENTS

Reviewer #1 (Remarks to the Author):

The manuscript describes a screen of small molecules modulating the inclusion of microexons. The authors use a dual luciferase reporter system containing a microexon. Depending on its splicing behavior (included or skipped), a different luciferase protein is generated. These reporters are used in two reciprocal configurations to minimize false positives resulting from an effect on the activity of one of the luciferase genes. As the results show, this is highly necessary, as for the majority of tested compounds with an effect on the ratio between the luciferase readouts this is due to an effect on the activity of one of the luciferase enzymes rather than the splicing event.

The authors use this reporter system to screen collections of small compounds and identify a set of molecules affecting the splicing behavior of the reporter. In general, with the exception of the positive controls, the effect sizes on the splicing reporter readout tend to be small. However, and somewhat surprisingly, they are detectable in RT-PCR validations, which should be much less sensitive than the luciferase assay.

In order to assess the specificity of the effect on microexons, the authors tried to quantify the effect of some small molecule hits on splicing and gene expression transcriptome-wide. This provides a rich dataset, but the analysis presented in the paper stays very much at the surface. The RNA-seq data could be analyzed more in depth, which might provide additional insights into the selectivity of the effect and enrich the paper as well as provide evidence for molecular targets (e.g. RBPs beyond Srrm4) of the small molecules. Which features of a splicing event determine whether it is affected by a small molecule? Weak consensus sequences? The presence of specific sequence motifs?

The authors provide such an analysis on the level of expression of RBPs, but if the claim is a selective effect on a subset of splicing events (all or some microexons), a splice site-centric analysis could be more convincing. At this point, the large number of RBPs showing changes in their gene expression levels rather sheds some doubt on the potential specificity of the effect to microexons.

In general, the manuscript presents an interesting screening strategy that is – for the most part – carefully controlled and described. The authors avoid overselling their results which makes the manuscript more accessible. However, - and maybe as a consequence of unclear results – the text is sometimes vague and therefore not too informative. While the principle of the approach and the identification of small molecules with a potentially preferential effect on microexons will certainly be of interest for researchers in the field, I am hesitant to endorse publication of the manuscript in its present form in a venue with a broad readership like Nature Communications.

We thank the reviewer for their positive comments and constructive feedback. As summarized in more detail below (also in Response to Reviewers 2 and 3) we have substantially revised our manuscript to further address the mechanism by which some of the compounds identified in our screen preferentially affect the inclusion levels of microexons relative to other classes of alternative splicing events. In response to the concerns raised above, we performed a systematic comparative analysis of cis-features surrounding splicing events impacted by compounds versus those that are not. While this highlighted a relatively weak branch site score as being a significant feature of exons that are modulated by the RDR00572 ‘activator’ compound, consistent with

microexons generally being associated with relatively weak 3' splice site/branch site scores ¹, a more important mechanism associated with the action of several compounds further analysed is that they affect the expression of specific trans-acting regulators that preferentially affect microexon splicing, including Srrm4 and Rbm38. These and additional data included in the revised manuscript are described in more detail below.

Specific points and suggestions:

We have numbered comments from Reviewer 1 for ease of reference when responding to similar comments from Reviewers 2 and 3.

1. Abstract, line 9: In light of the results presented in the paper, the word “selectively” is not accurate. The small molecules identified here do not appear to be selective for regulating microexon inclusion levels but to some extent affect other splicing events as well and expression levels of a substantial number of genes in general.

We have revised the manuscript to clarify in the abstract and elsewhere that some of the ‘hit’ compounds preferentially affect the splicing of microexons relative to other classes of exons.

2. Figure 1BC: There seems to be a systematic difference in PSI between the two configurations (B and C). The authors should comment on this.

We have added the following sentences in the manuscript to comment on this, but also to clarify that these shifts in PSI do not affect the linearity of the dose responsiveness of the reporters, page 5, Line 141: “Notably, for both Mef2d reporters, the insertion or deletion of a single nucleotide to generate the V1 and V2 configurations resulted in a consistent shift in PSI over the range of SRRM4 expression levels. Nonetheless, both configurations of each reporter displayed highly sensitive and linear responses to increasing concentrations of SRRM4”.

In addition, we now provide the modified sequences for the Mef2d microexons reporters (Figure S1A) and Shank2 microexon reporters (Figure S2A) to improve clarity to the reader.

3. Figure 2A: Putative activators and inhibitors with the strongest effect in one configuration typically do not show any effect in the other configurations. Why are they still considered candidates?

To address the reviewer’s concern, we have revised our description of hit selection criteria, and our method of plotting the screen data in Figure 2A, to better reflect the final criteria used to select compounds for further testing. Figure 2A plots B-scores of V1 and V2 reporter luminescence ratios the hit calling was based on. In the revised Figure 2A, we show hyperbolae fit to establish hit selection thresholds and that exclude the large majority of false positives, which exert opposite or unidirectional changes in luminescence ratios (see main text and Methods for details). Using this approach, approximately 0.8% of the tested compounds were initially scored as putative activators, and 0.7% of compounds were scored as putative inhibitors, whereas most of the remaining compounds resulted in a shift in B-score (see main text and Methods for details) in the opposite direction for the two reporters, indicating a change in reporter expression, and were considered false-positives, or else resulted in no change in B-score. Using the criterion, the majority of putative hits indicated in Figure 2A exert a similar effect-size on both V1 and V2

reporters. However, a small proportion of putative hits (close to the selection threshold) exerted a strong effect on one reporter while having a smaller effect on the other reporter.

4. In addition, the effect size of many “putative inhibitors” or “activators” is close to 0. Was there no cut-off on effect size? What is considered a significant effect?

We have clarified in the main text and Methods how we selected putative activators and inhibitors. To initially maximize the number of compounds for further testing following the primary screen, we fitted hyperbolae on the plot of B-scores of luminescence ratios (revised Figure 2A) to exclude false positives, while capturing compounds that induced similar effects on both V1 and V2 reporters. We have now updated the Methods section to more clearly define how the custom hyperbolae used as a hit selection threshold were generated:

“Hit selection criterion

Putative hits were selected by plotting the ratios of B-scores for Firefly and Nano luminescence for each reciprocal reporter (Figure 2A). We next utilized custom hyperbolae to generate a hit selection threshold, in order to exclude the majority of false positive generated from unidirectional changes in luminescence. Custom hyperbolae were generated according to the following formula:

$$(x - y/-7) * (-y + x/-7)/-2 = 4$$

where x and y are B-score of (FireflyV1/NanoV1) and B-score of (NanoV2/FireflyV2), respectively.

Using this hit selection criteria, we identified 738 (0.8%) small molecules as putative activators, and 644 (0.7%) small molecules as putative inhibitors.”

5. In addition, the use of semi-quantitative RT-PCR validations for such small effect sizes is questionable? Validation with qPCR should be provided for at least some of the candidates (across the range of effect sizes that the authors deem relevant).

We respectfully disagree with this comment. In our current study and in numerous previous studies (eg. {Irimia, 2014 #7900; Tapial, 2017 #7937; Han, 2017 #7939; Han, 2022 #8132}) we have collectively performed thousands of semi-quantitative RT-PCR assays to measure PSI values of alternative exons across diverse cell/tissue types and conditions, including those that have minimal PSI differences as detected by RNA-Seq analysis. In our experience these measurements are generally within the linear range of detection, and overall correlation rates for Δ PSI measurements between semi-quantitative RT-PCR and RNA-Seq assays are typically in the range of $r \sim 0.8$. Similarly, from the set of approximately 450 RT-PCR reactions performed for ~ 90 putative hit compounds, we observed an overall correlation of 0.80 between Δ PSI measurements from these assays and the B-score NLuc/RLuc ratio changes from the V1 and V2 reporter assays. This is shown in a new scatterplot in Figure 2D.

To better visualize the correlation between these measurements, and in response to a request from Reviewer 2, we have also included a heatmap and scatterplot comparing luminescence ratio B-score changes from the V1 and V2 reporter data with changes in PSI from the RT-PCR experiments (Figure 2C-D). We have also added a new Supplementary Table (S3) containing all Δ PSI values from the RT-PCR validations performed in this study.

6. Figure 2C: Why do both readouts (Firefly and Nano) increase in the left panel (or stay more or less unchanged in the case of Nano) while in the right panel one increases as the other one decreases (which seems to be the expected behavior if I understand the readout correctly)?

To clarify, these panels were included to confirm changes in expression of NLuc and FLuc in response to representative activator and repressor compounds, over the dilution series tested in the secondary screen, for the V1 reporter only. Consistent with the activator compound SAHA/Vorinostat promoting splicing of the *Mef2d* microexon, we observe an increase in expression of FLuc relative to NLuc, within the concentration range used in the screen. However, this compound may also promote the expression of both NLuc and FLuc transcripts from this reporter, such that it does not show a reciprocal decrease in NLuc expression. Conversely, for the inhibitor compound AW00693, we observe an increase in NLuc and simultaneous decrease in FLuc expression over its active concentration range. We have made changes to the main text to clarify the description of these panels.

Page 7, Line 205: *Notably, as expected AW00693 treatment resulted in a corresponding increased expression of Nano luminescence and decreased expression of Firefly luminescence (Figure 2B, upper panel), whereas treatment with SAHA/Vorinostat led to an increase in Firefly luminescence but not a reciprocal decrease in Nano luminescence, possibly due to this compound promoting both expression or stability and splicing of the reporter transcripts (Figure 2B, lower panel).*

7. Figure 2D: Why were some only tested with activators and others only with repressors? And especially in these cases, the directionality of the effect is often lost (activators having a repressive effect and vice versa), including for microexons. These results are not sufficiently explained and discussed.

In addition to assaying several alternative splicing events (*Mef2d*, *Mast2* and *Zdhhc20*) for responses to all tested activator and inhibitor compounds, we additionally assayed the *Ergic3* microexon in response to activators since it has a relatively low basal PSI in N2A cells under control conditions and therefore affords a wider response-range for measuring increased PSI. Similarly, for inhibitors, we assayed the *Trappc9* microexon since it has a relatively high basal PSI level in N2A cells under control conditions and therefore a larger response-range for measuring decreased inclusion. Notably, the expected directionality of PSI changes was observed for the majority (310/455; 68.0%) of the tested compounds in the RT-PCR validation experiments, with an exception being the *Ergic3* microexon, as some of the putative activators reduced its inclusion, potentially because of indirect effects. We have revised the manuscript text to better explain the basis for selecting specific alternative splicing events for validation assays, and we have also expanded the description of these results.

Page 8, Line 229:

*The alternative splicing events selected for RT-PCR validation include both microexons (≤ 27 nt) (i.e. *Trappc9*, *Ergic3* and *Mast2*) and longer (> 27 nt) neural-regulated exons (i.e. *Asph*, *Mrgn1*, and *Zdhhc20*). The *Ergic3* microexon was additionally selected for validation of putative activators since it has relatively low inclusion levels in N2A cells under control conditions. Conversely, the *Trappc9* microexon was selected for validation of putative inhibitors since it has relatively high inclusion in N2A cells under control conditions. This selection of test exons thus afforded an initial assessment of potential selectivity of the small molecule effects on microexons versus longer neural exons, as well as for their impact on microexons regulated through *Srrm4*-dependent versus -independent mechanisms.*

Overall, we detected a strong correlation ($r=0.75$) between the changes in FLuc/NLuc ratios from the primary screen and changes in PSI (Δ PSI) measured from the RT-PCR assays of Mef2d microexon inclusion levels from the V1 and V2 reporters (Figure 2D, mean Pearson's $r=0.80$). Moreover, most compounds resulted in splicing level changes of endogenous microexons in the same direction as the Mef2d reporter exon; however, several activator compounds reduced inclusion levels of the Ergic3 microexon, presumably reflecting that different microexons have distinct factor requirements and mechanisms of splicing (Figure 2C).

8. Lines 214-217: "This selection of test exons afforded an initial assessment of potential selectivity of the small molecule effects on microexons versus longer neural exons, and for their impact on microexons regulated through Srrm4-dependent versus -independent mechanisms." – This sentence is very vague. In light of the results this is understandable, but the statement could still be rephrased to convey a clearer message.

We have rewritten this section of the Results to better explain the multiple criteria used for selection of alternative splicing events for RT-PCR validation experiments (see above response to comment 7).

9. Line 219: 39 top-ranked compounds: what was the cutoff used? Why the top 39?

We have rephrased this description to improve clarity and indicate that we curated a set of representative hits from the top-ranked (i.e., largest mean Δ PSI for Srrm4-dependent microexons) compounds.

Page 9, Line 247: Following RT-PCR validation, a Tanimoto similarity index was used to cluster representative compounds with activity in microexon activation and repression, based on their 2D structural similarity (Figure 3A). Among the highest-ranking (i.e., largest mean Δ PSI for Srrm4-dependent microexons, Supplementary Table 2) activators of microexon splicing are...

We now include our ranking, as well as the raw Δ PSI data for all RT-PCR validations, as a new Supplementary Table (S3).

10. Figure 3A by itself is not very informative, maybe highlighting the groups presented in 3B or annotating other relevant clusters could add to the graph.

We have highlighted the compounds shown in Figure 3B in the Figure 3A heatmap, and known targets are indicated on the top x-axis.

11. Figure 4C: There seems to be a problem with the normalization of the RNA-seq counts. CPM was used, so the logFC should still be distributed around 0 (which is not the case for panels 2, 3 and 5).

Thank you for pointing this out. We have corrected this error; logFC change is now distributed around zero.

12. Figure 5BC: What is the proposed mechanism of action of these drugs? This is not a consequence of aberrant microexon splicing (if anything it might be a cause of the observed effect, given that Srrm4 is one of the targets). How do they affect expression of splicing regulators such as Srrm4?

In the revised manuscript we have further explored the mechanism by which some of the compounds affect splicing, also in response to comments from Reviewers 2 and 3. For example, we have demonstrated that microexon inhibitor compounds shown in Figures 5B and 5C, including AW00693 and T5985871, significantly reduce mRNA transcript levels of the key microexon regulator SRRM4 (qPCR assay, Figure 5C). We have now further confirmed this effect at the protein level by performing western blotting experiments (Figure 6B). Additionally, we have observed that both microexon inhibitor compounds promote the expression of RBM38 transcripts, which we recently identified as a major negative regulator of microexon inclusion {Han, 2022 #8132}. This was detected in our analysis of compound-treated N2A cells by RNA-Seq data analysis (Volcano plots focused on all annotated RBPs in Figure 6A), was further confirmed at the protein level for compound AW00693 (Figure 6B), and in an analysis of the overlap of effects of compound treatment with those of Rbm38 knockdown (Figure 6C).

Reviewer #2 (Remarks to the Author):

In this manuscript, Best AJ et al. screened small molecules to identify compounds that regulate micro exons. They employed dual Nano and Firefly luciferase alternative splicing minigenes, modeling screening reporters previously used by various labs (Hagiwara, Cooper, Black). The authors found that HDAC inhibitors and PKI inhibitors were the top hits. They performed RNA-seq of cell lines treated with several selected drugs to determine the breath of gene expression and splicing changes. The results indicated that these drugs affected expression of SRRM3/4. The authors knocked down SRRM4 in N2a cells and showed that this abolished the drug effect on a couple of micro exons.

The manuscript's strength lies in the scale of the screening effort, which involved testing ~95k small molecules. However, there is insufficient novelty nor new mechanistic insight in the findings, and the authors need to temper their conclusions. For instance, the HDAC inhibitors are clearly not "selective" activators of micro exons. SRRM3/4 are already well known to control micro exons, and the data supporting their involvement in the HDAC inhibitors' activity is relatively weak.

The authors position the manuscript for translational impacts by highlighting the roles of micro exons in developmental disorders. However, no data implicates the translatability of the identified hits. The analyses stop short at cell lines that are irrelevant to a disease setting. The manuscript also lacks sufficient details and statistical analyses to assess the data quality.

We thank the reviewer for their constructive feedback. As also mentioned in response to comments from Reviewers 1 and 3, we have made substantial changes to the manuscript to provide additional evidence that a subset of the screen-identified compounds preferentially impact the splicing of microexons relative to other classes of alternative splicing events (and relative to gene expression changes), additional insight into the mechanisms of selected compounds, and evidence of the potential for one of the activator compounds to rescue microexon splicing deficiency *in vivo*.

Major points

We have numbered comments from Reviewer 2 for ease of reference when responding to similar comments from Reviewers 1 and 3.

1. The authors can enhance the comprehensibility of the screen results for readers. For example, it is unclear why different groups of exons were tested for different drug groups (fig 2d). For benchmarking, the same exon groups should be tested unbiasedly for these selected drug groups.

We have rewritten this section of the manuscript to more clearly explain the basis for selection of specific alternative splicing events for validation purposes. Please also see response to comment 7 from Reviewer 1 for details.

2. The authors can easily acquire data for every drug-exon combination since the drug-treated samples and the splicing assay are available. This is important to evaluate the consistency of the impact of each of 156 drugs on the microexons vs longer exons. The authors should present the result in a more informative manner (e.g., a heatmap and others) than a simple violin plot in fig 2d. In addition, for transparency, the authors should release the PSI values of the tested exons upon drug treatment from the RT-PCR experiments, e.g., in a supplementary excel file.

We thank the reviewer for the excellent suggestion. We have included a heatmap comparing B-score changes from the V1 and V2 small molecule reporter data with changes in PSI from the RT-PCR experiments (Figure 2C and 2D). Also as requested, we added a Supplementary Table (S3) containing all Δ PSI values from the RT-PCR validation experiments performed in our study, and our rankings based on mean Δ PSI for Srrm4-dependent microexons.

3. The paper and the screen aim to identify small molecules that affect micro exon splicing, as the title and abstract suggest. However, the paper has yet to demonstrate the specificity of candidate drugs in affecting micro exons vs longer alternative exons. The RNA-seq data actually show they affect both.

As also mentioned in Response to Reviewer 1, we have rewritten parts of the manuscript to clarify and describe additional evidence showing that a subset of the screen-identified compounds preferentially affect the splicing of microexons relative to longer alternative exons. For example, this is apparent for the PKI Trametinib, and the two microexon inhibitors AW00693 and T5985871 compounds, which affect higher proportions of microexons than other classes of alternative splicing events, likely in large part because they significantly affect levels of Srrm4 and Rbm38 proteins, factors demonstrated previously to preferentially promote and repress microexon splicing, respectively (see Figure 6).

4. Moreover, each HDAC inhibitor affects expression of 10-35% genes and the splicing of 10-20% of micro exons. The authors have not presented convincing evidence to indicate specific effects of HDAC inhibitors on micro exon splicing vs global transcriptomic changes.

If we interpret the reviewer's question correctly, they are wondering whether the effect of the HDAC inhibitors on microexon splicing might be explained by coupled changes in expression of the corresponding genes? We have performed a new analysis to determine the degree of overlap between genes with altered microexon splicing, and genes that have altered expression upon HDAC inhibitor treatment. We detect no significant overlap for any of the HDAC inhibitors (please see Supplemental Figure S5C). Conversely, we observe that genes containing microexons affected by BRD6688 are less likely to change in expression upon HDAC inhibitor treatment than would be expected by chance. This suggests HDAC inhibitor-induced activation of microexon splicing is largely independent of expression changes involving the same gene.

5. The potential novelty of the manuscript is the identification of PKI inhibitors, but the authors have not followed up to provide mechanistic insights.

We have included new western blotting data in the manuscript showing that treatment with the PKI Trametinib leads to significant up-regulation of Srrm4 (a major positive regulator of microexons) and down-regulation of Rbm38 (a major negative regulator of microexons) at the protein level, whereas all other tested microexon regulators (i.e. Srsf11, PTB, Qki and Rbfox2) were not significantly affected. Using published iCLIP data^{2,3}, we further show that proximal intronic sequences upstream of exons that show increased inclusion upon treatment with Trametinib are significantly enriched in Srrm3/4 iCLIP binding. Taken together, these data suggest that treatment with Trametinib leads to an up-regulation of Srrm4, which then binds and regulates Trametinib-responsive exons. These results are now described on pages 13 and 14 of the revised manuscript and shown in Figures 6B and 6D.

6. The authors indicate that the micro exons are important in the brain. Therefore, the positive hits should be tested in the brain or neurons to evaluate their effects on micro exons.

To address this comment, we performed an in vivo microexon rescue experiment. Srrm4 haploinsufficient mice, which show broad (partial) skipping of neuronal microexons and ASD-like phenotypes, were systemically treated with HDACi Vorinostat/SAHA, or the orphan compound RDR00572. Early-stage postnatal mice were injected subcutaneously with either compound and sacrificed at postnatal day 8 for brain dissection and RNA isolation from whole cerebral cortex. RT-PCR assays were performed to determine the efficacy of the compounds in rescuing microexon inclusion levels. While we did not observe microexon rescue with SAHA/Vorinostat (data not shown), treatment with RDR00572 remarkably led to significant rescue of splicing levels (i.e. comparable to WT levels) for several analysed Srrm4-dependent microexons. These results are now shown in a new figure (Figure 7A and 7B) and described on pages 14 and 15 of the revised manuscript.

7. It is necessary to specify in the main text which nucleotide is added to and deleted from the mef2d microexon and at what position. This would help readers to understand the design. An important question is to what extent these modifications affect the basal splicing outcome and the screening results. These should be tested by RT-PCR in comparison to the minigene and discussed in detail, given the authors' attempt to present the values of their methods.

We have provided additional information to the manuscript to describe which nucleotides were added or deleted from the Mef2d (Figure S1A) and Shank2 (Figure S2A) microexons to generate V1 and V2 reporters. As also mentioned in response to Reviewer 1 (see response to comment 2), we have tested the effect of these mutations by RT-PCR to assess their impact on basal splicing levels. While they shift basal splicing levels (eg. by approximately +14 PSI (V1) and +5 PSI (V2) (for the Mef2d microexon) we have confirmed that they do not impact the sensitivity or linearity of the response of the reporters to the compound treatments. This is shown in Figures 1B and 1C and now clarified on page 5 of the revised manuscript, which now reads: *“Notably, for both reporters, the insertion or deletion of a single nucleotide to generate the V1 and V2 configurations resulted in a consistent shift in PSI over the range of SRRM4 expression levels. Nonetheless, both*

configurations of each reporter displayed highly sensitive and linear responses to increasing concentrations of SRRM4”.

8. To improve readers’ understanding of the minigene system, the authors should explain why it does not respond to SRRM4 when expressed from the CMV or PGK promoter? Additionally, the authors should specify which promoters were used for the single luminescence constructs in fig S1b?

We have added the following explanation on page 6: Initially, we observed that the luminescence ratios emitted from cell lines expressing the Mef2d splicing reporter from constitutively active promoters (CMV, PGK or EFS) were minimally responsive to increasing concentrations of SRRM4, likely due to high levels of expression of luciferase prior to induction of SRRM4 expression (data not shown).

9. Fig1, it would be helpful for readers to know whether the Nluc sequence included in both the inclusion isoform (V1) and the V2 exclusion isoform. The authors should also present Western blot data to show the 3xflag-tagged firefly protein is expressed as the expected size, to ensure its levels and activities reflect the designated splicing pathway other than some unknown processing of the minigene.

To address this comment, we have performed a western blot analysis using anti-FLAG antibody to confirm that the Mef2d-luciferase fusion proteins expressed from the V1 and V2 reporter minigenes are of the expected molecular weights, and also that their expression levels reciprocally shift in ratio in response to Srrm4 expression.

These data are now shown in Figure S1B and described on page 6; Further supporting that the detected changes in luminescence ratios are due to alternative splicing, we confirmed reciprocal changes in the ratios of protein isoforms expressed from these reporters in response to expression of SRRM4, by western blotting with anti-Flag antibody (Figure S1B).

We note that the changes in ratios of the protein isoforms in response to Srrm4 expression are not as linear as those observed by RT-PCR, likely because addition of the PEST degradation sequences to the vector ORFs, which target the reporter proteins for rapid turnover.

10. Fig1b-c: since NLuc and FLuc are separate assays producing different output units, their relative comparison based on these arbitrary units can be swung easily in either direction. The authors should explain how they achieved such high linearity and clarify why the other reporters do not appear as linear.

We have revised the manuscript text to explain in more detail how measurements of the relative ratios of NLuc and FLuc emission were used to infer splicing levels, and why these measurements achieved high linearity for the Mef2d reporter under control of the dox inducible partial CMV promoter, in contrast to the stronger promoters tested (see page 5). We have also clarified (page 6) why we think the Shank2 dual luciferase microexon reporter was not as linear; we believe this may in part be due to usage of an alternative 3’ splice site that produces an additional in-frame protein, which could impact protein stability (indicated by an asterisk on RT-PCR gel image and now shown in Figure S2E).

11. Fig 1e: it is unclear what statistics and cutoffs were used to define positive hits.

Lien 222 “Among the highest-ranking activators of microexon splicing are several”. Line 239 “The highest-ranking inhibitors of microexon splicing are also drug-like molecules” where can readers find the ranking information of drugs?

We have clarified the criteria used to define putative hits in Figures 1E and 2A in the main text and Methods (see also response to Reviewer 1, comment 4). We now define the formulae used to plot activity of the reporters as B-scores of ratios, as well as the formula used to generate custom hyperbolae that define hit selection threshold.

We have additionally provided our specific definition of ‘high-ranking’ throughout the manuscript (i.e. largest mean Δ PSI for Srrm4-dependent microexons), and the corresponding values in additional Supplementary Table (S3).

12. Line 247 “To globally assess the effect of small molecules on alternative splicing (and gene expression), we selected 27 top representative...” the selection criteria were mentioned but it is unclear how they were applied. The authors should clarify this information in the manuscript so readers understand if there is any selection bias. Treatment conditions are also unclear.

The 27 small molecules analyzed by RNA-Seq were manually selected based on results from the RT-PCR validation data, such that they represent a range of different classes of compounds, including HDAC inhibitors, PKIs, as well as compounds representing different structural groups without known targets. We have added these details to the manuscript (on page 10). We have also clarified which compound concentrations were used for the RNA-seq analysis (Figure S4A, and Supplementary Table S4) and for all experiments throughout the manuscript, as well as the timepoint used in this analysis (i.e. 24 hours) (page 10).

Fig 5, the authors should include data and images of the cell health and density at the time of rna collection as a quality control for the splicing and gene expression analysis.

We have included bright-field microscopy images, and an assessment of cell viability using propidium iodide (PI) staining and flow cytometry, at the time of RNA collection (Supplementary Figures S3A, S3B and S3C). For compounds BRD6688, CI-994, Trametinib and M98D13, we detected little if any effects on cell health and viability in N2A cells. The microexon inhibitor compounds AW00693 and T5985871 did however result in a partial reduction in cell proliferation and viability. These observations are now mentioned on page 10 of the manuscript.

13. Fig 6, protein expression and activity are often uncorrelated with their mRNA expression. The authors should test protein expression of these RBPs.

As requested, we have tested protein expression of several relevant RBPs following compound treatments. Western blot analysis confirmed that treatment with the PKI Trametinib leads to a significant increase in Srrm4 protein levels, and conversely a decrease in Rbm38 protein levels, while the two microexon inhibitors AW00693 and T5985871 resulted in pronounced reductions in Srrm4 protein levels (as well as other RBPs including Qki and Rbfox2). We did not detect a statistically significant increase in Srrm4 protein expression upon treatment with the HDAC inhibitors BRD6688 and CI-994. These results are shown in Figure 6B in the revised manuscript and are described on page 14. (See also response to comment 12 from Reviewer 1).

14. Fig6, the RNA-seq data do not appear to have biological replicates and statistics?

The RNA-Seq analysis of the four HDAC inhibitors (SAHA/Vorinostat, BRD6688, BRD4884 and CI-994) was performed using a total of two biological replicates. The RNA-Seq analysis for the remaining small molecule treatments was performed from single samples, and DMSO-treated control samples were analyzed in biological triplicate. The Δ PSI measurements measured between biological replicates generally were highly correlated, providing confidence in measurements from these data. We have now clarified these details in the Methods section.

15. Fig6c, validation using a few SRRM4 targets is underpowered, especially when these exons are different from those used in other figures. The authors should use RNA-seq to provide unbiased results.

We have repeated this experiment in order to include Trametinib treatment (Figure 6E). We show that for specific microexons (i.e. Lphn2 and Zmynd8), the effect of Trametinib is markedly reduced in a cell line expressing an shRNA targeting Srrm4. Taken together with the new western blot data (Figure 6B) revealing an increase in Srrm4 protein levels upon Trametinib treatment, and the enrichment of Srrm3/4 iCLIP reads directly upstream of Trametinib-activated microexons (Figure 6D), these data suggest that Trametinib leads to an up-regulation of SRRM4, which binds upstream and promotes the splicing of microexons affected by treatment with this compound (page 14).

16. Are other (or most) HDAC inhibitors identified as hits in the screen?

Yes, multiple HDAC inhibitors were identified as hits in the screen. We highlight nine different HDAC inhibitors in Figure 1E that promote splicing inclusion in the V1 and V2 Mef2d microexon reporters. We included eight different HDACis (i.e. Trichostatin A, LAQ824, JNJ-26481585, Givinostat, Panobinostat, Vorinostat/SAHA, SB939 and Scriptaid) in our systematic RT-PCR validation of screen hits (Figure 2C and Supplementary Table S3). We also show an RNA-Seq analysis of treatments for four different HDAC inhibitors: SAHA/Vorinostat, BRD6688, BRD4884 and CI-994.

Minor points

17. Fig S2b-c: what are the * isoforms?

We performed Sanger sequencing of the * isoforms. This revealed that they are generated by an alternative 3' splice site. We have mentioned this finding in the manuscript (Figure S2E) and also how it likely confounds the linearity of the Shank reporter (page 5) and response to previous comment.

18. statistics is missing in fig 5a, 6c

The RT-PCR data in Figure 5A were generated using single biological samples per compound treatment. Importantly however, similar effects on microexon splicing inhibition were confirmed for all four inhibitor compounds tested. Furthermore, adding confidence to these results, the effects of the inhibitor compounds measured by RT-PCR assays following treatment of the human cell line (NCI-82) in Figure 5A are highly consistent with the results from the RNA-Seq analyses of treated N2A cells (Figure 4B, right hand panel), in which ~75% of all Srrm3/4-dependent microexons showed reduced inclusion levels.

The experiment for Figure 6C (now Figure 6E) was repeated to include Trametinib treatment, and is a representative experiment from three biological replicates.

Reviewer #3 (Remarks to the Author):

Summary:

The authors establish and demonstrate a reciprocal dual luciferase reporter system for screening libraries of molecules for splice-modifying activities. They use this reporter system to screen a large library of small molecules for compounds that promote or inhibit microexon inclusion. They validate representative hits from this screen using various techniques including RNA-seq, qPCR, minigene assays, and dose-response measurements. Several of the compounds identified as promoting microexon inclusion are histone deacetylase (HDAC) inhibitors. The paper proposes that HDAC inhibitors promote microexon inclusion by up-regulating Srrm4 and down-regulating Ptbp2, and that compounds that inhibit microexon inclusion do so by down-regulating Srrm4 and Srrm3.

There is a great deal of interest in screens for splice-modifying drugs, and I believe the reporter system described here is an important contribution to those screening efforts. In particular, Figure 2B convincingly shows that that, had the authors used a non-reciprocal screening system, they would have identified many false-positive hits (i.e., drugs that affect the activity of one or both luciferase readouts, but do not affect splicing of the reporter transcript).

I believe that the core contribution of this paper—the reciprocal dual reporter system—is thoughtful, rigorous, and has the potential to substantially impact screening efforts to identify splice-modifying drugs. However, the data provided to support the mechanistic hypothesis of how identified compounds function are less convincing. The authors may wish to focus more on the screening aspect of their manuscript, and soften their statements about mechanism.

We thank the reviewer for the positive comments and helpful feedback. As also requested by Reviewers 1 and 2, we have added new data and substantially revised the manuscript text to provide a more detailed description of technical aspects of the screen, additional validation of the screen results, and more insight into how selected compounds preferentially affect the splicing of microexons over other classes of alternative splicing.

Major comments:

1. The mechanisms proposed in Figure 6D, and discussed in 370-374, seem overly speculative. The changes in expression observed by RNA-seq are suggestive, but they do not on their own show mechanism. The data in Figure 6C does show that CI-994 activity was abolished by Srrm4 knockdown, which provides evidence that CI-994 acts through an Srrm4-dependent mechanism. But BRD6688 activity was abolished by Srrm4 knockdown at only 2 of the 4 loci assayed. I also do not see evidence that the HDAC inhibitors are functioning by down-regulating Ptbp2 expression, or that any of the microexon inhibitor compounds are functioning by down-regulating Srrm3 or Srrm4 expression. Note that HDAC inhibitors have recently been shown to regulate splicing by reducing H3K9me2 and increasing H3K9ac at the affected exon (PMID: 35688133). Is it possible that the identified HDAC inhibitors are regulating microexons by a similar mechanism?

We agree that the previous Figure 6D was overly speculative and therefore have removed it. As also summarized above in response to comment 12 from Reviewer 1, and comments 5 and 13 from Reviewer 2, we have provided the following additional data and associated text revisions in the revised manuscript to expand on the proposed mechanisms by which specific compounds activate or repress microexon splicing:

- We show that treatment with the PKI Trametinib leads to a significant up-regulation of Srrm4 and down-regulation of Rbm38 protein levels (Figure 6B). Furthermore, microexons which increase in PSI upon Trametinib treatment are enriched in Srrm3/4 iCLIP reads directly upstream of the affected exons (Figure 6D). Finally, for specific events (i.e. Lphn2 and Zymnd8) the effect of Trametinib on microexon inclusion is substantially reduced in N2A cells expressing an shRNA targeting Srrm4 (Figure 6E). These results are described on pages 13 & 14.
- We also demonstrate that the microexon inhibitor compounds AW00693 and T5985871 lead to a marked reduction in SRRM4 protein levels (Figure 6B), which likely contributes to the broad and substantial reduction in Srrm4-dependent microexons. These results are described on page 13.

We have also clarified that the BRD6688 HDAC inhibitor may promote microexon splicing through both Srrm4 dependent and independent mechanisms. While we agree that the recently described effect of HDAC inhibitors on reducing H3K9me2 and increasing H3K9ac levels around SMN2 exon 7 is interesting, we believe such a mechanism is less likely to be responsible for the broad stimulatory effects on microexons observed in the present study. At the effective concentrations of these compounds used in the present study, they have relatively widespread effects on gene expression, increasing the likelihood that they affect splicing levels more indirectly by altering levels of RBPs. This point is made in the revised Discussion (pages 15-17).

2. The manuscript lacks clarity on whether biological or technical replicates were performed for some of the experiments. To avoid confusion, this information should be explicitly mentioned in the main text or figure legends. For instance, Figures 5A, 5C, and 6A-C do not contain information about biological or technical replicates. Moreover, uncertainties should be reported for the PSI measurements in Figures 5A and 6C, gene expression measurements in Figures 5C and 6A, and Δ PSI measurements in Figure 6B.

We have updated the manuscript in order to clarify whether biological and/or technical replicates were performed, as well the numbers of replicates performed for each experiment.

While the RT-PCR data in Figure 5A were generated from a single experiment, it is important to note that all four compounds tested displayed similar and substantial relative levels of splicing inhibition on seven analyzed microexons. Furthermore, the results from the RT-PCR analysis in Figure 5A, which was performed using the human cell line NCI-82, are highly consistent with results from the RNA-seq analysis of the inhibitor compounds performed using N2A cells. Similarly, in Figure 5C, while these data were also generated from a single experiment with three technical replicates, the effects of the small molecule treatments were highly consistent across three independent cell lines and at the three different test concentrations. We believe that the extent of validation and resulting high level of consistency of the effects of the compounds observed across multiple independent alternative splicing events, different cell lines, and concentrations, provides strong support for the robustness and reproducibility of our findings.

We have also clarified in the Methods section that in Figure 6A, the RNA-seq data for CI-994 treatment was generated from two biological replicates, while RNA-seq data for the RDR00572, Trametinib, AW00693 and T5985871 treatments were generated from single treatments. Furthermore, the experiment for previous Figure 6C (now Figure 6E) was repeated in order to include the PKI Trametinib and these data were generated from three biological replicates.

3. Some of the data visualizations might be improved.

i. For Figure 5C, instead of the bar plot and points, it might be better to show separate titration curves for each microexon.

Figure 5C shows quantification of mRNA expression (qRT-PCR) for three different genes (SYP, SRRM4, SRRM3) across three different cell lines. We tried plotting these data in different ways and found that barplots make it easier to appreciate the quantitative differences between the inhibitory effects of the compounds as well as the remarkably low degree of variance in the measurements between the three different cell lines tested. We have clarified the description of Figure 5C in the manuscript on page 13.

ii. For Figure 6A, volcano plots having specific genes of interested highlighted would better illustrate the fold-change and FDR.

We thank the reviewer for this suggestion. We have replaced Figure 6A with volcano plots indicating fold change and FDR for RBPs (as changes for all genes are provided in Figure 4E) , and highlighting RBPs that have significant changes in mRNA expression following compound treatments.

iii. For Figure 6B, the data for each of the four microexons assayed would be better displayed separately, and error bars should be shown for each microexon/treatment combination. Statements about additivity and synergy should be supported by quantitative comparisons of Δ PSI values under different treatments.

The previous Figure 6B was generated from a single experiment and therefore we could not generate error bars for the splicing measurements for individual microexons. However, instead, we have provided a barplot that affords a direct quantitative comparison of Δ PSI measurements for the four assayed microexon across the different compound treatments. This reveals an overall good degree of consistency between the relative effects of the compound treatments on different microexons. These data have been moved to Supplemental Figure 6D to free-up space for new mechanistic data in the main Figure 6.

4. It would be good to quantify protein levels by western blot to more definitively establish the gene expression changes observed in Figure 6A that are argued to contribute to drug mechanism.

As requested, also by Reviewers 1 and 2, we performed western blotting experiments to confirm changes in protein expression of relevant RBPs following compound treatments. Western blot analysis confirmed that treatment with the PKI Trametinib leads to a significant increase in Srrm4 protein levels, and that the two microexon inhibitors AW00693 and T5985871 markedly reduced Srrm4 protein levels (as well as other RBPs including Qki and Rbfox2). We did not detect a statistically significant increase in Srrm4 protein expression upon treatment with the HDAC

inhibitors BRD6688 and CI-994. We have now incorporated and discussed these results in the revised manuscript (see Figure 6B and pages 13 & 14).

Minor comments:

1. The authors should mention that this screening system can be applied to screens that do not involve small molecules, e.g., to ASO libraries or to CRISPR libraries.

Thank you for this suggestion. We have added this important point to the Discussion.

2. It might make sense to cite this study, which has discussed regulation of microexons by proteins involved in chromatin organization: Genome-wide CRISPR-Cas9 Interrogation of Splicing Networks Reveals a Mechanism for Recognition of Autism-Misregulated Neuronal Microexons, PMID: 30388412.

We have now referenced this study in the Discussion.

3. Gene names should be italicized.

We have italicized gene names.

4. In Figure 2D, it might make sense to group the microexons by size, and note in the plot which are Srrm4-dependent.

Following requests from Reviewers 1 and 2, we have replaced the previous Figure 2D with a heatmap that displays the Δ PSI measurements from the RT-PCR assays, and the corresponding B-score changes from the V1 and V2 reporter assays (see Figure 2C). This Figure also indicates which test exons are Srrm4 dependent and their length. We have also provided a new Supplementary Table (S3) for all Δ PSI values determined by RT-PCR.

5. In Figure 2D, it is unclear why so many RT-PCR validated activators inhibit splicing of the Ergic3 microexon, and why some RT-PCR validated inhibitors activate splicing in the other microexons. Please clarify in the main text.

We have updated the text to comment on this observation. Please also refer to our response to comment 7 from Reviewer 1.

6. Did the identified drugs affect cell viability? Were there any differences in cell morphology or proliferation rate? One might expect this would happen with small molecule drugs at micromolar concentrations, and it might be relevant for mechanism.

Also in response to comments from Reviewers 1 and 2, we have provided representative bright-field microscopy images, an assessment of cell viability (propidium iodide staining and flow cytometry), and an assessment of cell proliferation rate (EdU staining) at the time of RNA collection (see Supplementary Figure 3). For compounds BRD6688, CI-994, Trametinib and M98D13, we detect minimal effects on cell health and viability. However, for the microexon inhibitor compounds AW00693 and T5985871, we observe a reduction in cell proliferation and viability. Treatment with the HDAC inhibitor CI-994 and the PKI Trametinib significantly reduced

cell proliferation without an apparent effect on cell viability. These observations have are described on page 10 of the revised manuscript.

7. In line 321 there is a typo; I suggest the correction “expression the REST4 isoform” -> “expression of the REST4 isoform”.

We have corrected this typo.

8. The phrase “whereas the effects of...” in line 372 is confusing because the microexon inhibitors are also proposed to acct in an Srrm4-dependent manner.

We have re-written this section of the text to more accurately describe the results, and to represent the more recent mechanistic data included in the manuscript.

9. In supplemental Figure 2, “*” needs to be explained in the figure legend as a nonspecific product. It might make sense to sequence this product and report what it is.

We have performed Sanger sequencing of the * isoform, which reveals that it is produced by an in-frame alternative 3’ splice site (Figure S2E), which may contribute to the reduced linearity of the Shank2 reporter relative to the Mef2d reporter. We now mention this on page 5.

REFERENCES

1. Raj, B. *et al.* A global regulatory mechanism for activating an exon network required for neurogenesis. *Mol Cell* **56**, 90-103 (2014).
2. Gonatopoulos-Pournatzis, T. *et al.* Autism-Misregulated eIF4G Microexons Control Synaptic Translation and Higher Order Cognitive Functions. *Mol Cell* **77**, 1176-1192 e16 (2020).
3. Gonatopoulos-Pournatzis, T. *et al.* Genome-wide CRISPR-Cas9 Interrogation of Splicing Networks Reveals a Mechanism for Recognition of Autism-Misregulated Neuronal Microexons. *Mol Cell* **72**, 510-524 e12 (2018).

REVIEWER COMMENTS

Reviewer #1 (Remarks to the Author):

In the revised manuscript, the authors toned down the claim of "selectivity" of the effect for microexons and added mechanistic insights and technical controls and clarifications. They have addressed all my comments in a thorough and convincing manner.

Reviewer #2 (Remarks to the Author):

The authors have responded to a majority of previous concerns. Several items related to biological replicates and experimental rigor remained unaddressed.

Fig 7. To make RDR00572 rescue of micro exons move convincing, please use RNA-seq, which is a straightforward experiment and provides a global unbiased picture instead of four cherry-picked candidates. Four candidate exons (a panel different from those in other figures) is not sufficiently convincing.

The authors have not addressed the original comment "validation using a few SRRM4 targets is underpowered, especially when these exons are different from those used in other figures. The authors should use RNA-seq to provide unbiased results." please note that this is not to question the results of other experiments but the rigor of this experiment itself. Therefore, the authors' argument is not convincing by using other experimental results without directly addressing the concern of the bias confirmation.

Replicate number is not mentioned for fig5a. Please be consistent and transparent about biological replicates for all experiments and make sure at the very minimum $n=3$ (maybe except RNA-seq). The authors stated in the rebuttal letter (not manuscript) that RT-PCR data in Figure 5A were generated using single biological samples per compound treatment. At least three independent biological replicates are needed regardless of results from other experiments.

The authors should be transparent and mention in the main text when RNA-seq experiments have only a single or two biological replicates.

Please release the PSI values, not just the Δ PSI values from the RT-PCR validation experiments.

Reviewer #3 (Remarks to the Author):

Please see attached PDF, which contains the text of the review.

Referee report for "High-throughput sensitive screening of small molecule modulators of microexon alternative splicing using dual Nano and Firefly luciferase reporters" by Best et al. (revision 1).

5 February 2024

I thank the authors for their responsiveness to reviewer critiques. I think the manuscript is much improved and, over all, is a valuable contribution to the study of splice-modifying drugs.

I have one remaining major critique: the relevance of Fig. 6E, as well as the logic behind the authors' interpretation of Fig. 6E, is unclear:

- If a drug acts by changing the expression level of Srrm4, one would not (contrary to what the main

text suggests) expect to see an epistatic effect with shRNA KD of Srrm4. For example, if drug treatment increases Srrm4 expression by 2x in the absence of shRNA, there is no reason to suspect that the drug would not increase Srrm4 expression by approximately 2x in the presence of shRNA. I therefore do not understand the relevance of claims of epistatic effects in Fig. 6E.

- Claims in the main text regarding Fig. 6E are not clearly supported by the data. Part of the problem is that the figure shows PSI values, not the Δ PSI values that readers need in order to assess the claims in the main text. But having worked out these Δ PSI values myself, it is still not clear to me that the claims made in the main text regarding epistatic effects on Δ PSI values are supported.
- If the mechanisms of any drugs were attributable to drug effects on Srrm4 expression, Δ PSI values should track Srrm4 protein levels upon drug treatment, shRNA KD of Srrm4, or both. Western blot data for Srrm4 for each of the samples in Fig. 6E would be needed to establish that this is, in fact, happening. In particular, such Western data would also be needed to distinguish whether observed epistatic effects are due to epistasis in the effect that drug treatment and shRNA treatment have on Srrm4, or due to some other cause.

I suggest that the authors reconsider how Fig. 6E is presented and interpreted. I think other data (especially in Fig. 6B and 6D) already provides a sufficient starting point for future mechanistic studies, and that the authors should not have to further track down drug mechanisms. But I also think it's important that the authors not make unsupported statements about the potential mechanisms of the drugs they identify, as this could negatively affect subsequent work.

The rest of my comments pertain to minor issues with the presentation and are only meant to help the authors improve the paper:

1. I suggest showing gene names in splicing gels and other figures in italics and all caps, as is the convention (Figs. 1B, 1F, 5A, 6E, 7B, S2C, S5D)
2. I suggest showing the protein names in western blots in camel case, as is the convention (Fig. 6B).
3. I suggest labeling the colorbar in Fig. 2C with the term "B-score" to make clear which quantity is being plotted.
4. I suggest citing a paper describing the B-score in the main text of the paper. This is a statistic that is very specific to high-throughput screens and many readers might not be familiar with it. One reference I found to be useful is Brideau et al. (2003) in the Journal of Biomolecular Screening.
5. I suggest adjusting the colorbar ticks in Fig. 3A; right now it looks like the maximum value isn't 1.0, and that the minimum value isn't 0.0.
6. I suggest adjusting the colorbar ticks in Fig. 6C; right now the scale looks neither linear nor log.
7. I suggest re-rendering the y-axis labels in Fig. 6C; these are slightly clipped.
8. I suggest labeling the iCLIP data in Fig. 6D as being for Srrm3/4, both in the figure and legend.
9. I suggest also showing ramps for the different drug concentrations in Fig. 5C.
10. I suggest showing Δ PSI values in Fig. 6E, perhaps even instead of raw PSI values, as readers need to know the Δ PSI values to assess the claims made in the main text.

11. I suggest double-checking the comparisons in Fig. 7B. Some comparisons labeled n.s. seem to be highly significant by eye.

12. In line 435-437, it is unclear which "previous reports" the authors are referring to. If these reports refer to ref. 27, this should be clarified. If not, references should be given for these previous reports.

13. Given the samples they have, the authors may wish to address the question in order to shed further light on potential drug mechanisms: in the samples for Fig. 7B, does drug treatment increase the expression of Ssrn4 or other RBPs, either at the mRNA or protein level?

RESPONSE TO REVIEWERS

We thank the reviewers for their thorough review of our revised manuscript and their helpful and constructive final comments.

To summarize, Reviewer 1 is satisfied with our revisions and supportive of publication. Reviewer 3 is also supportive of publication and asked us to provide some text clarifications and minor changes, as well as improvements to some of the figures. Reviewer 2, while largely satisfied with the revisions, requested two new experiments, which we have performed.

Below, we summarize the revisions to our manuscript to address the comments.

Comments from each reviewer are in bold italicized text, followed by our response

Reviewer #1:

In the revised manuscript, the authors toned down the claim of "selectivity" of the effect for microexons and added mechanistic insights and technical controls and clarifications. They have addressed all my comments in a thorough and convincing manner.

Reviewer #2:

The authors have responded to a majority of previous concerns. Several items related to biological replicates and experimental rigor remained unaddressed.

Fig 7. To make RDR00572 rescue of micro exons move convincing, please use RNA-seq, which is a straightforward experiment and provides a global unbiased picture instead of four cherry-picked candidates. Four candidate exons (a panel different from those in other figures) is not sufficiently convincing. The authors have not addressed the original comment "validation using a few SRRM4 targets is underpowered, especially when these exons are different from those used in other figures. The authors should use RNA-seq to provide unbiased results." please note that this is not to question the results of other experiments but the rigor of this experiment itself. Therefore, the authors' argument is not convincing by using other experimental results without directly addressing the concern of the bias confirmation.

We previously performed the requested RNA-Seq analysis but decided not to describe the findings in the manuscript because the results were inconclusive. We detected a relatively small number of modest differences in microexon splicing levels in mice treated with the RDR00572 compound. This is not entirely surprising given that the RNA-seq analysis was performed using whole cerebral cortex samples, which comprise a highly heterogeneous population of cells. To achieve better sensitivity, we would ultimately need to perform deeper coverage RNA-Seq using full-length single cell/nuclear RNA-Seq profiling to capture changes specifically in neuronal populations. Such an analysis which would be non-trivial and time consuming, and therefore we feel that it is beyond the scope of our current study.

Nevertheless, to address the reviewer's comment, we have revised the text to clarify we are not making any claims about the selectivity of the compound *in vivo*, but instead are demonstrating feasibility of subcutaneous delivery of a small molecule to achieve statistically significant rescue of Srrm4-dependent microexons within the cerebral cortex. To our knowledge, this is the first demonstration that subcutaneous administration of a small molecule can rescue brain disorder-associated changes in microexon splicing in the brain and we therefore feel that it adds significant interest to the manuscript.

Replicate number is not mentioned for fig5a. Please be consistent and transparent about biological replicates for all experiments and make sure at the very minimum n=3 (maybe except RNA-seq). The authors stated in the rebuttal letter (not manuscript) that RT-PCR data in Figure 5A were generated using single biological samples per compound treatment. At least three independent biological replicates are needed regardless of results from other experiments.

The goal of this experiment was to assess using semi-quantitative RT-PCR assays whether compound treatments affect microexon splicing, and whether such changes can be interpreted in terms of changes in the expression of SRRM4 and/or SRRM3. The main take home message was that from a single biological replicate experiment we were able to observe highly consistent effects of the compound treatments across multiple different test exons, and that these changes were also consistent with negative effects of the same compounds on both SRRM4 and SRRM3 expression levels. However, to address the reviewer's comment, we have repeated the experiment in Figure 5A three more times and now provide quantification of the compound-induced PSI changes from all four replicate experiments, normalized to the DMSO-only treatment. These results are shown in the new (right) panel in Figure 5A and confirm our initial conclusions.

The authors should be transparent and mention in the main text when RNA-seq experiments have only a single or two biological replicates.

We have taken care to correct any remaining omissions regarding replicate numbers in the revised manuscript. Where relevant, numbers of replicates are indicated in the figure legends.

Please release the PSI values, not just the Δ PSI values from the RT-PCR validation experiments.

We have included all PSI values (in addition to Δ PSI values) as an additional sheet in Supplementary Table 3.

Reviewer #3:

I thank the authors for their responsiveness to reviewer critiques. I think the manuscript is much improved and, over all, is a valuable contribution to the study of splice-modifying drugs.

I have one remaining major critique: the relevance of Fig. 6E, as well as the logic behind the authors' interpretation of Fig. 6E, is unclear:

- ***If a drug acts by changing the expression level of Srrm4, one would not (contrary to what the main text suggests) expect to see an epistatic effect with shRNA KD of Srrm4. For example, if drug treatment increases Srrm4 expression by 2x in the absence of shRNA, there is no reason to suspect that the drug would not increase Srrm4 expression by approximately 2x in the presence of shRNA. I therefore do not understand the relevance of claims of epistatic effects in Fig. 6E.***

- ***Claims in the main text regarding Fig. 6E are not clearly supported by the data. Part of the problem is that the figure shows PSI values, not the Δ PSI values that readers need in order to assess the claims in the main text. But having worked out these Δ PSI values myself, it is still not clear to me that the claims made in the main text regarding epistatic effects on Δ PSI values are supported.***

- ***If the mechanisms of any drugs were attributable to drug effects on Srrm4 expression, Δ PSI values should track Srrm4 protein levels upon drug treatment, shRNA KD of Srrm4, or both. Western blot data for Srrm4 for each of the samples in Fig. 6E would be needed to establish that this is, in fact, happening. In particular, such Western data would also be needed to distinguish whether observed epistatic effects are due to epistasis in the effect that drug treatment and shRNA treatment have on Srrm4, or due to some other cause.***

I suggest that the authors reconsider how Fig. 6E is presented and interpreted. I think other data (especially in Fig. 6B and 6D) already provides a sufficient starting point for future mechanistic studies, and that the authors should not have to further track down drug mechanisms. But I also think it's important that the authors not make unsupported statements about the potential mechanisms of the drugs they identify, as this could negatively affect subsequent work.

We thank the reviewer for this helpful feedback. In our opinion the clearest result from Figure 6E is that the large positive effect of RDR00572 treatment on microexon splicing is not diminished in shSrrm4-expressing cells. Together with the observations that Srrm4 transcript and protein levels are unaffected by RDR00572 treatment, and that the magnitude of the effect of treatment with the compound vs Srrm4 depletion are uncoupled across events, this suggests that the MOA of RDR00572 is independent of Srrm4.

In the revised manuscript we have clarified our interpretations of the data previously shown in Figure 6. As requested, we have also edited the figure to provide dPSI values, rather than PSI values. Given the challenges in interpreting these data, we have moved Figure 6E to the Supplemental Information (now Supplementary Figure 6B).

The rest of my comments pertain to minor issues with the presentation and are only meant to help the authors improve the paper:

1. ***I suggest showing gene names in splicing gels and other figures in italics and all caps, as is the convention (Figs. 1B, 1F, 5A, 6E, 7B, S2C, S5D)***

Gene names have been changed to follow the accepted convention in all figures, i.e. italicised and all-caps for human, and capital-first letter followed by lower case for mouse.

2. I suggest showing the protein names in western blots in camel case, as is the convention (Fig. 6B).

Please see response to 1.

3. I suggest labeling the colorbar in Fig. 2C with the term “B-score” to make clear which quantity is being plotted.

We have added the term “B-score” to Figure 2C in order to improve clarity.

4. I suggest citing a paper describing the B-score in the main text of the paper. This is a statistic that is very specific to high-throughput screens and many readers might not be familiar with it. One reference I found to be useful is Brideau et al. (2003) in the Journal of Biomolecular Screening.

We thank the reviewer for pointing out this reference, which we have now cited.

5. I suggest adjusting the colorbar ticks in Fig. 3A; right now it looks like the maximum value isn't 1.0, and that the minimum value isn't 0.0.

We have updated the colour bar in Figure 3A.

6. I suggest adjusting the colorbar ticks in Fig. 6C; right now the scale looks neither linear nor log.

Fig. 6C does not include a colour bar, but the reviewer may be referring to Fig. 5B. We have now updated the colour bar in this figure.

7. I suggest re-rendering the y-axis labels in Fig. 6C; these are slightly clipped.

We thank the reviewer for spotting this error and have re-rendered the figure.

8. I suggest labeling the iCLIP data in Fig. 6D as being for Srrm3/4, both in the figure and legend.

Done.

9. I suggest also showing ramps for the different drug concentrations in Fig. 5C.

Thank you for this suggestion. We have now added ramps to Figure 5C.

10. I suggest showing Δ PSI values in Fig. 6E, perhaps even instead of raw PSI values, as readers need to know the Δ PSI values to assess the claims made in the main text.

We have followed the reviewer's suggestion and included Δ PSI values in this figure. This panel has now been moved to Fig. S6A.

11. I suggest double-checking the comparisons in Fig. 7B. Some comparisons labeled n.s. seem to be highly significant by eye.

The figure is correct. Some comparisons, while reproducible, did not yield a p-value < 0.05 as there was insufficient statistical power with two replicates and correction for multiple testing.

12. In line 435-437, it is unclear which "previous reports" the authors are referring to. If these reports refer to ref. 27, this should be clarified. If not, references should be given for these previous reports.

We have included two appropriate references.

13. Given the samples they have, the authors may wish to address the question in order to shed further light on potential drug mechanisms: in the samples for Fig. 7B, does drug treatment increase the expression of Ssrn4 or other RBPs, either at the mRNA or protein level?

While we agree with the reviewer that it will be interesting to determine the mechanism of action of RDR00572 in neurons, to adequately address this question would require a dedicated study employing additional approaches that we feel is beyond the scope of the current study. Please also see our response to Reviewer 2.

We thank the reviewers again for their helpful comments.

REVIEWERS' COMMENTS

Reviewer #2 (Remarks to the Author):

I disagree with the authors' interpretation of the inconclusive RNA-seq results. The figure 7 legend states that whole cerebral cortex samples were used. Cell heterogeneity is therefore not a differentiating factor. The authors have not described whether the RNA-seq result supported the RT-PCR results in fig 7 or whether they are consistent.

The inclusion of a very small group of microexons, distinct from those in other figures and highly selected for fig 7, does not provide adequate statistical power to support authors' conclusion and statement. For instance, the assertion in the abstract "One of these compounds rescues the splicing of microexons in the cerebral cortex of an autism mouse model haploinsufficient for *Srrm4*" could be overstated.

My intention was to advocate for greater transparency and scientific rigor in publishing. While I acknowledge that dissecting cell type specificity is beyond the scope, it's crucial that the authors accurately represent the results, especially since the data do not robustly support the drug rescue hypothesis. Given that they already know the drug rescue is not as robust as well supported by the RNA-seq results, the authors should state the RNA-seq results as is and temper their conclusions and statements accordingly, reserving stronger assertions for future studies with supporting data.

It is important to avoid making unsupported statements, particularly in the realm of drug and therapeutic studies. Caution should be taken in drawing definitive conclusions from preliminary or inconclusive results.

Reviewer #3 (Remarks to the Author):

I thank the authors for addressing my last critique. I remain unconvinced of the mechanistic interpretation of the data in Supplemental Fig. 6, but I also think this mechanistic interpretation is not an essential part of the authors' work. I would therefore be happy to see this valuable paper published.

Point-by-point response to the reviewers' comments

REVIEWERS' COMMENTS

Reviewer #2 (Remarks to the Author):

I disagree with the authors' interpretation of the inconclusive RNA-seq results. The figure 7 legend states that whole cerebral cortex samples were used. Cell heterogeneity is therefore not a differentiating factor. The authors have not described whether the RNA-seq result supported the RT-PCR results in fig 7 or whether they are consistent.

The inclusion of a very small group of microexons, distinct from those in other figures and highly selected for fig 7, does not provide adequate statistical power to support authors' conclusion and statement. For instance, the assertion in the abstract "One of these compounds rescues the splicing of microexons in the cerebral cortex of an autism mouse model haploinsufficient for *Srrm4*" could be overstated.

My intention was to advocate for greater transparency and scientific rigor in publishing. While I acknowledge that dissecting cell type specificity is beyond the scope, it's crucial that the authors accurately represent the results, especially since the data do not robustly support the drug rescue hypothesis. Given that they already know the drug rescue is not as robust as well supported by the RNA-seq results, the authors should state the RNA-seq results as is and temper their conclusions and statements accordingly, reserving stronger assertions for future studies with supporting data.

It is important to avoid making unsupported statements, particularly in the realm of drug and therapeutic studies. Caution should be taken in drawing definitive conclusions from preliminary or inconclusive results.

Response: We thank the reviewer for their critical feedback. To address the reviewer's concerns, in the previous submission version we toned down our assertions in the abstract and main text in order to more accurately reflect our observations. Following guidance from Dr. Minju Ha, we have made an additional change to the Abstract to clarify our results from the compound rescue experiments: "One of these compounds rescues the splicing of several analyzed microexons in the cerebral cortex of an autism mouse model haploinsufficient for *Srrm4*".

Reviewer #3 (Remarks to the Author):

I thank the authors for addressing my last critique. I remain unconvinced of the mechanistic interpretation of the data in Supplemental Fig. 6, but I also think this mechanistic interpretation is not an essential part of the authors' work. I would therefore be happy to see this valuable paper published.

Response: We thank the reviewer for helping to improve our manuscript and supporting its publication.